# Graph Neural Ricci Flow: Evolving Feature from a Curvature Perspective

**Jialong Chen, Bowen Deng, Zhen Wang, Chuan Chen**[*]**, Zibin Zheng**
Sun Yat-sen University
{chenjlong7, dengbw3}@mail2.sysu.edu.cn
{wangzh665, chenchuan, zhzibin}@mail.sysu.edu.cn

## Abstract

Differential equations provide a dynamical perspective for understanding and designing graph neural networks (GNNs). By generalizing the discrete Ricci flow (DRF) to attributed graphs, we can leverage a new paradigm for the evolution of node features with the help of curvature. We show that in the attributed graphs, DRF guarantees a vital property: The curvature of each edge concentrates toward zero over time. This property leads to two interesting consequences: 1) graph Dirichlet energy with bilateral bounds and 2) data-independent curvature decay rate. Based on these theoretical results, we propose the **G**raph **N**eural **R**icci **F**low (GNRF), a novel curvature-aware continuous-depth GNN. Compared to traditional curvature-based graph learning methods, GNRF is not limited to a specific curvature definition. It computes and adjusts time-varying curvature efficiently in linear time. We illustrate that GNRF performs excellently on diverse datasets.

## 1 Introduction

Graph Neural Networks (GNNs) have currently achieved significant success in tasks such as community detection (Liu et al., 2020), product recommendation (Wu et al., 2022; Gao et al., 2023) molecular design (Zhang et al., 2021; Wieder et al., 2020), and enhancing language models (Chen et al., 2024; Jin et al., 2023). One of the most successful ideas in designing GNNs is to stack several message-passing layers, allowing nodes to receive information and update their representations within multiple hops (Kipf & Welling, 2016; Veličković et al., 2017; Wu et al., 2019).

Recent research has revealed a close connection between these layered GNNs and differential equations (DEs). Oono & Suzuki (2019) first proposed the idea of viewing Graph Convolutional Networks (GCN) (Kipf & Welling, 2016) as discrete dynamical systems. While Chamberlain et al. (2021), using the heat diffusion equation, derives the continuous-depth counterpart for Graph Attention Networks (Veličković et al., 2017). By establishing DEs for node representations over time, more refined and theoretically sound evolution strategies can be designed, including energy conservation (Rusch et al., 2022), anti-symmetry (Gravina et al., 2022), and repulsion (Wang et al., 2022).

Most DE-GNNs are based on the heat equation and its variants (Chamberlain et al., 2021; Thorpe et al., 2022; Choi et al., 2023; Li et al., 2024; Bodnar et al., 2022). However, the classical heat equation forces the temperature in the system to become uniform over time, leading to a loss of expressive node representations in GNNs inspired by this equation when reaching an equilibrium state. In this paper, we break away from this fixed mindset for the first time and turn to explore the benefits of another important differential equation — the Ricci flow — for graph learning.

In differential geometry, Ricci flow is metaphorically described as a process where a complex manifold gradually becomes "regular". This process is governed by the Ricci curvature, causing regions with larger absolute curvature values to decay more significantly. When defining edge curvature based on node-attributed graphs, we find that the Ricci Flow works: it forces the edge curvature to concentrate towards zero quickly, thus time-efficiently yielding stable and non-smooth node representations.

---

[*]Corresponding author.

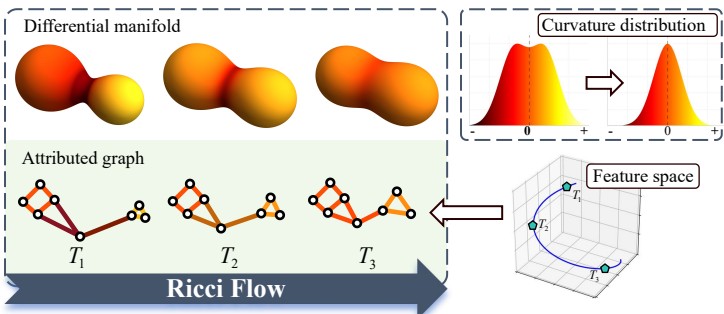

Figure 1: Analogize the feature evolution process on the node-attributed graph to the Ricci flow in differential manifolds, where the curvature gradually concentrates to zero.

Based on this observation, we design a novel continuous-depth GNN called the Graph Neural Ricci Flow (GNRF). To our knowledge, GNRF is the first deep graph learning model based on time-varying edge Ricci curvature. (SelfRGNN (Sun et al., 2022) is a potentially confusing related work, but they focus on curvature in the embedded space rather than on the edges.) Previously, the mainstream paradigm for utilizing Ricci curvature in graph learning was graph rewiring (Nguyen et al., 2023; Fesser & Weber, 2024; Shen et al., 2024), where edge curvature was considered an intrinsic property related only to topology, precomputed, and stored. Additionally, they consider only a specific curvature definition and require quadratic time complexity. GNRF, for the first time, defines time-varying edge curvature based on node attributes and introduces an auxiliary network to compute curvature for any given definition within linear time.

Our contributions can be summarized as follows:

1. We are the first to consider the evolution of node attributes from the perspective of attirbute discrete Ricci flow, providing theoretical guarantees on decay rate and Dirichlet energy.
2. We propose GNRF, the first GNN, to apply time-varying edge curvature and introduce an auxiliary network for unified and efficient curvature computation.
3. We empirically test how well the GNRF fits the theory and describe its mechanism. We also verify its significant validity on a wide range of datasets.

## 2 PRELIMINARIES

**Notations.** We consider a simple undirected graph $\mathcal{G} = (\mathcal{V}, \mathcal{E})$, where $\mathcal{V}$ is a node set with size of $|\mathcal{V}|$ and a edge set $\mathcal{E} \subseteq \mathcal{V} \times \mathcal{V}$ with size of $|\mathcal{E}|$. The $i$-th node is associated with a feature vector $\boldsymbol{x}_i \in \mathbb{R}^n$, and the matrix form of all features is denoted as $\boldsymbol{X} = [\boldsymbol{x}_i, \ldots, \boldsymbol{x}_N]^T \in \mathbb{R}^{N \times n}$. Similarly, let $\boldsymbol{H}(t) = [\boldsymbol{h}_i(t), \ldots, \boldsymbol{h}_N(t)]^T \in \mathbb{R}^{N \times m}$ be the node representations evolved to time $t$ according to a differential equation. The initial representation $\boldsymbol{H}(0)$ is obtained from $\boldsymbol{X}$ through a function $F$, i.e., $\boldsymbol{H}(0) = F(\boldsymbol{X})$. We denote the time-varying attribute on edge $i \sim j$ as $w_{ij}(t)$. $w$ may be defined by $\boldsymbol{H}$, in which case we have: $w_{ij}(t) = w(\boldsymbol{h}_i(t), \boldsymbol{h}_j(t))$. $\kappa_{ij}(t)$ represents the edge curvature under any given definition.

**Differential equation inspired GNNs (DE-GNNs).** Continuous depth is the fundamental characteristic that distinguishes DE-GNN from other GNN architectures. It can be unified as:

$$\frac{\partial \boldsymbol{H}(t)}{\partial t} = f(\mathcal{G}, \boldsymbol{H}(0), \boldsymbol{H}(t)), \quad t \in [0, T]. \tag{1}$$

The update function $f$ can be either non-parametric (Oono & Suzuki, 2019; Veličković et al., 2017; Wu et al., 2023) or parametric (Li et al., 2020; Xu et al., 2023). The heat diffusion equation is the most extensively discussed in DE-GNN methods (Chamberlain et al., 2021; Thorpe et al., 2022; Li et al., 2024). A classic description of it is provided by Chamberlain et al. (2021):

$$\frac{\partial \boldsymbol{h}_i(t)}{\partial t} = \text{div}[g \cdot \nabla \boldsymbol{h}_i(t)] = \sum_{j \sim i} a(\boldsymbol{h}_i(t), \boldsymbol{h}_j(t)) (\boldsymbol{h}_j(t) - \boldsymbol{h}_i(t)), \tag{2}$$

where $g$ is the diffusion flux, and $a(\cdot, \cdot)$ is the similarity function for representations which is usually assumed to be non-negative (Chamberlain et al., 2021; Li et al., 2024).

**Edge curvature on weighted graph.** For a weighted graph $\mathcal{G}$ where each edge $i \sim j$ corresponds to a weight $w_{ij}$, the edge curvature $\kappa_{ij}$ measures the tightness of the connection between the ego-graph of node $i$ and the ego-graph of node $j$. $\kappa_{ij}$ has multiple definitions, with the earliest being the purely combinatorial one proposed by Forman (2003) via CW complex, known as the Forman-Ricci curvature $\kappa_{ij}^{\mathrm{FR}}$. In addition, Ollivier (2007) proposes Ollivier-Ricci curvature $\kappa_{ij}^{\mathrm{OR}}$, which is based on the optimal transport distance between ego-graphs. Devriendt & Lambiotte (2022) proposes viewing a weighted graph as a resistance network and establishing the resistance curvature $\kappa_{ij}^{\mathrm{RC}}$ via effective resistance. See Appendix A.1 for a more detailed introduction.

Currently, curvature-based graph learning methods generally treat curvature as a static property of edges, using it to perform graph rewiring (Nguyen et al., 2023; Topping et al., 2021; Fesser & Weber, 2024), edge sampling (Liu et al., 2023), or neighbor reweighting (Li et al., 2022). Their underlying idea is similar: to mitigate the influence of edges with extreme positive/negative curvature. owever, these methods often precompute Ricci curvature, resulting in a quadratic time complexity. Moreover, they focus on using a specific definition of curvature, which may lack sufficient robustness when dealing with different graph data (Southern et al., 2023; Attali et al., 2024). In contrast, Our method provides a way to evolve node features in linear time to adjust curvature, which is applicable to any curvature definition.

**Discrete Ricci flow (DRF).** The Ricci flow was first introduced by (Hamilton, 1982) in differential geometry and was later extended to complex networks by (Ollivier, 2009; 2010) using the definition:

$$\frac{\partial w_{ij}(t)}{\partial t} = -\kappa_{ij}(t) w_{ij}(t), \quad w_{ij}(t) > 0, \tag{3}$$

which is referred to as the discrete Ricci flow. Recently, DRF has found its applications in network mining, such as in community detection (Ni et al., 2019; Lai et al., 2022), network alignment (Flow, 2018), and biological structures prediction (Baptista et al., 2024). However, they all focus on graphs without node attributes, where edge weights are treated as inherent properties, which is inconsistent with the majority of datasets used in modern graph deep learning.

## 3 APPLYING DRF ON ATTRIBUTED GRAPHS

We aim to establish a dynamical system on node-attributed graphs similar to Equation (3). The intuitive idea is to treat edge weights as a function of the attributes of the two connected nodes, i.e., $w_{ij}(t) = w(\boldsymbol{h}_i(t), \boldsymbol{h}_j(t))$, where $w(\cdot, \cdot) > 0$ :

$$\frac{\partial w(\boldsymbol{h}_i(t), \boldsymbol{h}_j(t))}{\partial t} = -\kappa_{ij}(t) w(\boldsymbol{h}_i(t), \boldsymbol{h}_j(t)). \tag{4}$$

We refer to this dynamical system as **Attribute Discrete Ricci Flow** (Attri-DRF). Note that curvature is defined based on edge weights, which themselves are functions of node attributes. This implies that in Attri-DRF, curvature is also determined by the node attributes. Consequently, as node attributes evolve over time, the curvature changes accordingly. Attri-DRF also, for the first time, realizes the modeling of graphs curvature in continuous-time scenario.

Ricci flow is analogous to heat diffusion of the metric on a Riemannian manifold. A fundamental characteristic of standard heat diffusion is that the temperature distribution of any heat field gradually becomes smoother over time, ultimately reaching a steady state (thermal equilibrium). This implies that GNNs based on heat diffusion (such as GRAND Chamberlain et al. (2021) and GRAND++ Thorpe et al. (2022)) will exhibit a similar steady state as time approaches infinity, where the node features gradually become uniform—consistent with the well-known over-smoothing phenomenon. By contrast, Ricci flow, as a 'diffusion on the metric', leads to a 'metric equilibrium', meaning that the curvature on the manifold gradually becomes uniform. Specifically, in the context of Attri-DRF, we can conclude the following:

**Lemma 1** (informal). *Consider the Attri-DRF on any edge $i \sim j$, if $w_{ij}(t)$ is monotonic over a non-zero interval $[t_1, t_2]$ and $|w_{ij}(t_2) - w_{ij}(t_1)|$ is sufficiently close to 0, then the average curvature over $[t_1, t_2]$, i.e., $\mathbb{E}_{t \in [t_1, t_2]}(|\kappa_{ij}(t)|)$, is also sufficiently close to 0.*

Lemma 1 indicates that if the Attri-DRF on an edge approaches an equilibrium state, the curvature of that edge must necessarily be close to zero. Since excessively large or small curvature can lead to over-smoothing or over-squashing (Nguyen et al., 2023), Attri-DRF offers a unified and novel solution to these issues: leveraging the evolution of the Ricci flow to drive the curvature towards 0. It is worth noting that this process evolves automatically over time and is independent of any specific definition of curvature, which contrasts sharply with the currently popular graph rewiring paradigm (Nguyen et al., 2023; Topping et al., 2021; Fesser & Weber, 2024; Shen et al., 2024).

To have a closer look, from now on, we let $w_{ij}(t) \equiv \cos(\boldsymbol{h}_i(t), \boldsymbol{h}_j(t)) + 1 + \epsilon$ be a non-negative cosine similarity, where $\epsilon$ is a small positive number. Additionally, let $|\boldsymbol{h}(t)| \equiv 1$ to prevent numerical vanishing or explosion. At this point, $w$ satisfies: $w \in [\epsilon, 2 + \epsilon]$.

## 3.1 DIRICHLET ENERGY

The Dirichlet energy $E(\boldsymbol{H}(t))$ of a graph $\mathcal{G}$ is used to characterize the smoothness of attributes between nodes. As neighboring node representations become similar, $E$ tends towards 0. Proving that $E$ has a lower bound is a common theoretical means to demonstrate the ability of GNN to resist over-smoothing (Wang et al., 2022; Zhou et al., 2021a). Using Theorem 2, we demonstrate that when Attri-DRF stabilizes (i.e., $\kappa(t)$ approaches 0), $E$ has both upper and lower bounds. This means that the Attri-DRF avoids over-smoothing and prevents neighboring nodes from having excessive differences. See Appendix A.2 for more details about Dirichlet energy.

> **Theorem 2** (Informal). *Consider the Attri-DRF with $w_{ij}(t) \equiv \cos(\boldsymbol{h}_i(t), \boldsymbol{h}_j(t)) + 1 + \epsilon$ and $|\boldsymbol{h}| \equiv 1$. If within $[t_1, t_2]$, each edge of graph $\mathcal{G}$ reaches evolutionary equilibrium, then at this time $\mathcal{G}$ has both non-trivial upper and lower bounds on Dirichlet energy that are independent of the definition of curvature.*

Here, we provide an intuitive explanation. Extreme smoothing results in completely uniform node attributes, whereby the edge curvature is determined solely by the local topology. In most graphs without additional assumptions, differences in local topology lead to different curvatures, which contradicts Lemma 1. Therefore, for almost all graphs, we can derive a nonzero lower bound for the Dirichlet energy.

## 3.2 UNIFORM CURVATURE DECAY

We have characterized the asymptotic equilibrium of Attri-DRF and how it benefits graph learning. Now, we further discuss how this equilibrium is achieved, with particular attention to its practical significance—specifically, whether it can reach a satisfactory state within a finite time.

> **Theorem 3** (Informal). *Consider the Attri-DRF with $w_{ij}(t) \equiv \cos(\boldsymbol{h}_i(t), \boldsymbol{h}_j(t)) + 1 + \epsilon$ and $|\boldsymbol{h}| \equiv 1$ on any edge $i \sim j$. Assume that the curvature is bi-Lipschitz continuous when considered as a function of the weights. For any arbitrarily small positive number $\delta$, if $|\kappa_{ij}(0)| > \delta$, then $\kappa_{ij}$ will first decay to a value smaller than $\delta$ at $t = \mathcal{O}\left(\ln\left(\delta^{-1}\right)\right)$.*

Theorem 3 implies that Attri-DRF requires about $\mathcal{O}(\ln \delta^{-1})$ time to achieve a steady state. There are three points worth noting: **(1)** This bound is practically feasible: for $\delta = 10^{-5}$, $\ln(\delta^{-1})$ remains no greater than 10. **(2)** The result is independent of $w_{ij}(0)$ and $\kappa_{ij}(0)$: for any given $\delta$, the result applies to every edge in any $\mathcal{G}$. In contrast, Newton's law of cooling indicates that the time for heat diffusion to reach a steady state depends on the initial conditions (Winterton, 1999). **(3)** The decay rate is uniform: Theorem 3 can be applied to all edges of the same graph, leading to a synchronized decay of all curvatures. This eliminates the need to balance the differences in the evolution processes of various edges. In summary, the feature evolution of Attri-DRF is feasible, independent of the initial state, and uniform.

# 4 OUR MODEL: GRAPH NEURAL RICCI FLOW

## 4.1 INCORPORATING ATTRI-DRF INTO THE DE-GNN FRAMEWORK

Attri-DRF alleviates the issue of node representation quality degradation caused by excessively high or low curvature, and shows advantages in evolution time. To leverage these beneficial properties,

we next demonstrate how to derive the general form of the DE-inspired GNN (Equation (1)) from Attri-DRF (Equation (86)).

By expanding Equation (86) using the chain rule, we obtain[1]:

$$\left\langle \frac{\partial w(\boldsymbol{h}_i, \boldsymbol{h}_j)}{\partial \boldsymbol{h}_i}, \frac{\partial \boldsymbol{h}_i(t)}{\partial t} \right\rangle + \left\langle \frac{\partial w(\boldsymbol{h}_i, \boldsymbol{h}_j)}{\partial \boldsymbol{h}_j}, \frac{\partial \boldsymbol{h}_j(t)}{\partial t} \right\rangle = -\kappa_{ij}(t) w(\boldsymbol{h}_i, \boldsymbol{h}_j). \tag{5}$$

Equation (5) splits the effect of Attri-DRF on $w_{ij}$ into two parts: the effect on $\boldsymbol{h}_i$ and the effect on $\boldsymbol{h}_j$. To weigh these two parts, we introduce a scaling function $\lambda(\boldsymbol{h}_i(t), \boldsymbol{h}_j(t))$, i.e., $\left\langle \frac{\partial w(\boldsymbol{h}_i, \boldsymbol{h}_j)}{\partial \boldsymbol{h}_i}, \frac{\partial \boldsymbol{h}_i(t)}{\partial t} \right\rangle = \lambda(\boldsymbol{h}_i(t), \boldsymbol{h}_j(t)) \left\langle \frac{\partial w(\boldsymbol{h}_i, \boldsymbol{h}_j)}{\partial \boldsymbol{h}_j}, \frac{\partial \boldsymbol{h}_j(t)}{\partial t} \right\rangle$. We can now focus solely on one side:

$$\left\langle \frac{\partial w(\boldsymbol{h}_i, \boldsymbol{h}_j)}{\partial \boldsymbol{h}_i}, \frac{\partial \boldsymbol{h}_i(t)}{\partial t} \right\rangle = -\frac{\kappa_{ij}(t) w(\boldsymbol{h}_i, \boldsymbol{h}_j)}{1 + \lambda(\boldsymbol{h}_i(t), \boldsymbol{h}_j(t))}. \tag{6}$$

Equation (6) provides the first constraint for $\boldsymbol{h}_i(t)$, while $|\boldsymbol{h}_i(t)| \equiv 1$ is the second one. Under the satisfaction constraint, we minimize $\|\partial_t \boldsymbol{h}_i(t)\|$, which means that $\boldsymbol{h}_i(t)$ always applies only the slightest change to satisfy the Attr-DRF, which guarantees that the evolution of $\boldsymbol{h}_i(t)$ is numerically stable. Formally, this leads us to the following optimization objective:

$$\min \left\| \frac{\partial \boldsymbol{h}_i(t)}{\partial t} \right\|, \quad \text{s.t. } (\textbf{I}) \left\langle \frac{\partial w(\boldsymbol{h}_i, \boldsymbol{h}_j)}{\partial \boldsymbol{h}_i}, \frac{\partial \boldsymbol{h}_i(t)}{\partial t} \right\rangle = -\frac{\kappa_{ij}(t) w(\boldsymbol{h}_i, \boldsymbol{h}_j)}{1 + \lambda(\boldsymbol{h}_i(t), \boldsymbol{h}_j(t))}, \quad (\textbf{II}) \; |\boldsymbol{h}_i(t)| \equiv 1. \tag{7}$$

**Proposition 4.** *The optimization objective given by Equation (7) has a closed-form solution with linear time and space complexity as follows:*

$$\frac{\partial \boldsymbol{h}_i(t)}{\partial t} = -\kappa'_{ij}(t) \left[ \boldsymbol{h}_j - \cos\left(\boldsymbol{h}_i, \boldsymbol{h}_j\right) \boldsymbol{h}_i \right], \tag{8}$$

*where* $\kappa'_{ij}(t) = \frac{\kappa_{ij}(t) w(\boldsymbol{h}_i, \boldsymbol{h}_j)}{(1 + \lambda(\boldsymbol{h}_i(t), \boldsymbol{h}_j(t)))(1 - (\boldsymbol{h}_i^T \boldsymbol{h}_j)^2)}$.

Considering that the update of $\boldsymbol{h}_i(t)$ is actually influenced by all the neighbors of the node $i$, denoted as $\mathcal{N}(i)$, we let $j$ in Equation (8) range over all elements in $\mathcal{N}(i)$ and take the sum. This leads to deriving a Ricci flow-based node representation evolution, which forms the DE-GNN model. We refer to this as Graph Neural Ricci Flow (GNRF):

$$\frac{\partial \boldsymbol{h}_i(t)}{\partial t} = \sum_{j \sim i} \underbrace{-\kappa'_{ij}(t)}_{\text{weight}} \left[ \boldsymbol{h}_j(t) - \underbrace{\cos\left(\boldsymbol{h}_j(t), \boldsymbol{h}_i(t)\right)}_{\text{damping factor}} \boldsymbol{h}_i(t) \right]. \tag{9}$$

By comparing Equation (9) with Equation (2), one can get more insights between GNRF and heat diffusion. In early graph heat diffusion models like GRAND (Chamberlain et al., 2021), the aggregation weight is always positive, which is considered the driving force behind the smoothing of node attributes (Wang et al., 2022). Negative weights, on the other hand, are considered repulsive forces, making node attributes dissimilar. In GNRF, the sign of the aggregation weights $(-\kappa')$ is opposite to that of the curvature $(\kappa)$, which results in nodes with positive curvature being pushed apart, while other nodes are pulled closer. As demonstrated by experiments, we find that this helps in generating smoother decision boundaries.

Another distinction between GNRF and heat diffusion is the damping factor. It restricts the degree of the feature evolution. An intuitive explanation is that when $\|\boldsymbol{h}(t)\| \equiv 1$, then:

$$\left\| \boldsymbol{h}_j(t) - \cos\left(\boldsymbol{h}_j(t), \boldsymbol{h}_i(t)\right) \boldsymbol{h}_i(t) \right\|^2 = 1 - \cos^2\left(\boldsymbol{h}_j(t), \boldsymbol{h}_i(t)\right). \tag{10}$$

so excessive similarity or dissimilarity between $\boldsymbol{h}_i$ and $\boldsymbol{h}_j$ (close to 1 or -1) may weaken $\|\partial_t \boldsymbol{h}_i(t)\|$.

## 4.2 UNIFYING CURVATURES VIA EDGENET

The key distinction of GNRF from other GNNs lies in its ability to perceive time-varying edge curvature. However, curvature is often computationally expensive. For instance, calculating the resistance curvature $\kappa^{\text{RC}}$ requires computing the matrix pseudo-inverse, while Ollivier-Ricci curvature

---

[1]When unambiguous, we omit the independent variable $t$ for simplicity.

$\kappa^{\mathrm{OR}}$ involves solving the optimal transport distance. These complexities make real-time curvature computation a major bottleneck, hindering the broader application of curvature.

GNRF addresses this challenge by introducing an auxiliary network. Recent research has shown that the Wasserstein distance can be approximated in linear time using a simple network (Chen & Wang, 2024). Inspired by this, we aim to leverage the universal approximation capability of neural networks to seek a general solution for approximating the curvature under any definition.

> **Theorem 5** (Informal). *There exists a unified network structure called **EdgeNet**, which takes as input the weights of all edges connected to nodes $i$ and $j$, and approximates $\kappa'_{ij}$ with arbitrary precision in linear time, i.e., $\kappa'_{ij}(t) = \mathrm{EdgeNet}(\{w_{ik}(t)|k \sim i\}, \{w_{jk}(t)|k \sim j\})$. The definition of edge curvature can be Forman-Ricci curvature $\kappa^{\mathrm{FR}}_{ij}(t)$, Ollivier-Ricci curvature $\kappa^{\mathrm{OR}}_{ij}(t)$ and approximate resistance curvature $\widetilde{\kappa}^{\mathrm{RC}}_{ij}(t)$.*

EdgeNet not only enables the computation of curvature in linear time, but more importantly, it overcomes the limitations of existing definitions by realizing adaptive curvature. Despite the variety of curvature definitions, to the best of our knowledge, there is no theoretical foundation that definitively identifies one as superior. Moreover, based on experimental results in Southern et al. (2023) and Attali et al. (2024), we observe that different curvature definitions have a significant impact on the effectiveness of graph learning, yet it remains challenging to establish clear empirical guidelines. Therefore, using adaptive curvature may be a more ideal approach, and our experimental results support this view.

**Computational complexity.** Following the calculation protocol in Blakely et al. (2021), the computational complexity of GNRF is $\mathcal{O}(l|\mathcal{V}|n^2 + l|\mathcal{E}|n)$, where $l$ is the number of iteration steps of ODE and $n$ is the feature length. This complexity is linear with graph size ($|\mathcal{E}|$ and $|\mathcal{V}|$) and consistent with $l$-layer GCN. The method of pre-computing curvature usually requires square complexity, such as $\mathcal{O}(|\mathcal{V}|^2)$ (FOSR (Karhadkar et al., 2022)).

**Differential Equation Solver.** We use the Adams-Moulton method implemented by `torchdiffeq` (Chen et al., 2018) as the default solver for GNRF. Although GNRF performs well on most solvers, the Adams-Moulton method often achieves numerically stable solutions with larger fixed step sizes.

## 5 EXPERIMENT

**Datasets.** To evaluate the model fairly, we collect a total of 14 datasets from 6 commonly used node classification benchmarks. We report 12 of these datasets in the main experiment: Cornell, Wisconsin, and Texas from WebKB used in Pei et al. (2020); Roman-Empire, Tolokers, Amazon-ratings, Minesweeper and Questions from Heterophilous Graph benchmark (Platonov et al., 2023); Cora_Full, Cora_ML, DBLP and Pubmed from CitationFull benchmark (Bojchevski & Günnemann, 2017). In addition, to verify the scalability of the model, we also introduce two larger-scale data sets: OBGN-Arxiv from Open Graph Benchmark (Hu et al., 2020) and OGBN-Year from Lim et al. (2021). We report the performance and cost of models on these two datasets in scalability experiments. For all datasets, we uniformly adopted a random split strategy of 60%/20%/20% for the training, validation, and test sets. We report the mean and standard deviation of the experiments based on ten different splits.

**Comparison method.** We compared GNRF with two categories of methods. The first category is discrete-depth GNNs, including two classic models: Graph Convolution Network (Kipf & Welling, 2016) and Graph Attention Network (Veličković et al., 2017), as well as two advanced state-of-the-art models: Feature Selection GNN (Maurya et al., 2022) and Directed GNN (Rossi et al., 2024). Recent studies show that simple modifications can significantly improve classic model performance (Luo et al., 2024; Platonov et al., 2023). Therefore, we add residual connections to GCN and GAT, resulting in enhanced versions: GCN+res and GAT+res. The second category is continuous-depth GNNs, including Graph Neural Diffusion (Chamberlain et al., 2021), GRAND++ (Thorpe et al., 2022), Allen-Cahn Message Passing (Wang et al., 2022) and High-order Graph Diffusion Network (Li et al., 2024). To evaluate the advantages of adaptive curvature, we also compared two variants: $\mathrm{GNRF}_{FRC}$ and $\mathrm{GNRF}_{ARC}$. Instead of using EdgeNet, these variants directly use the definitions of Forman-Ricci curvature and approximate resistance curvature to obtain $\kappa$.

## 5.1 SEMI-SUPERVISED NODE CLASSIFICATION

| Hom. level
# Node | Corn.
0.1227
183 | Wisc.
0.1778
251 | Texas
0.0609
183 | R. Emp.
0.0000
22,662 | Tolo.
0.6344
11,758 | Mine.
0.6827
10,000 | Ques.
0.8359
48,921 | A.-rat.
0.3803
24,492 | C._Full
0.5670
19,793 | PubM.
0.8024
19,717 | DBLP
0.8279
17,716 | C._ML
0.7885
2,995 |
|---|---|---|---|---|---|---|---|---|---|---|---|---|
| **Discrete-depth GNNs** | | | | | | | | | | | | |
| GCN | 55.14
(±8.46) | 61.60
(±7.00) | 60.00
(±6.45) | 71.23
(±0.22) | 79.61
(±0.66) | 74.79
(±1.78) | 50.21
(±2.24) | 37.99
(±0.61) | 68.06
(±0.98) | 86.74
(±0.47) | 83.93
(±0.34) | 87.07
(±1.21) |
| GCN+res | 70.11
(±10.21) | 69.50
(±6.00) | 71.66
(±4.13) | 73.91
(±0.66) | 83.44
(±0.61) | 90.13
(±0.70) | 75.45
(±2.31) | 48.17
(±0.55) | 69.53
(±0.44) | 86.91
(±0.31) | 82.64
(±0.51) | 85.62
(±0.72) |
| GAT | 53.64
(±11.1) | 60.00
(±11.0) | 61.21
(±8.17) | 77.40
(±1.53) | 81.45
(±0.92) | 80.12
(±1.11) | 65.47
(±0.88) | 42.52
(±1.22) | 67.55
(±1.23) | 87.24
(±0.55) | 80.61
(±1.21) | 84.12
(±0.55) |
| GAT+res | 65.42
(±7.33) | 72.20
(±4.00) | 73.45
(±6.11) | 81.55
(±0.26) | 83.91
(±0.33) | 92.45
(±0.77) | 76.95
(±0.85) | 50.00
(±0.43) | 67.33
(±0.68) | 87.50
(±0.40) | 83.51
(±0.72) | 85.11
(±0.19) |
| DirGNN | 76.51
(±6.14) | 80.50
(±5.50) | 76.25
(±6.31) | 85.21
(±0.44) | 82.64
(±0.75) | 81.52
(±0.41) | 59.95
(±0.79) | 46.66
(±0.61) | 67.80
(±0.53) | 86.94
(±0.55) | 81.22
(±0.54) | 85.66
(±0.31) |
| FSGNN | 87.43
(±3.65) | 87.60
(±5.10) | 85.15
(±3.91) | 83.64
(±0.71) | 81.01
(±0.65) | 85.53
(±0.41) | 71.41
(±0.32) | 40.02
(±0.51) | 71.90
(±0.65) | 90.24
(±0.71) | 83.31
(±0.55) | 89.44
(±0.43) |
| **Continuous-depth GNNs** | | | | | | | | | | | | |
| GRAND | 81.76
(±13.9) | 84.00
(±7.50) | 81.70
(±8.42) | 60.12
(±0.75) | 79.01
(±0.45) | 80.56
(±3.12) | 54.90
(±2.12) | 37.53
(±0.36) | 67.66
(±1.01) | 86.79
(±0.57) | 84.60
(±0.99) | 88.49
(±0.81) |
| GRAND++ | 81.34
(±7.12) | 81.50
(±6.00) | 79.34
(±7.22) | 68.13
(±0.51) | 78.85
(±0.56) | 78.55
(±2.11) | 60.14
(±0.88) | 38.01
(±0.50) | 67.53
(±0.74) | 87.21
(±0.33) | 85.21
(±0.24) | 88.44
(±0.53) |
| ACMP | 85.66
(±5.10) | 86.50
(±5.00) | 87.65
(±3.54) | OOM | OOM | 85.11
(±1.06) | 71.92
(±0.55) | 37.32
(±0.64) | OOM | 88.01
(±1.44) | 82.31
(±0.44) | 76.11
(±2.12) |
| HiD-Net | 83.53
(±7.10) | 81.30
(±6.60) | 77.11
(±6.91) | 62.37
(±0.73) | 79.50
(±0.71) | 84.33
(±0.65) | 63.77
(±0.85) | 41.19
(±1.03) | 68.11
(±0.64) | 88.60
(±0.45) | 84.92
(±0.31) | 89.00
(±0.51) |
| **GNRF** | 87.28
(±3.12) | 88.00
(±2.00) | 87.39
(±4.13) | 86.25
(±0.46) | 83.96
(±0.39) | 95.03
(±0.20) | 73.86
(±1.18) | 46.89
(±1.08) | 72.12
(±0.50) | 90.37
(±0.76) | 85.73
(±0.76) | 89.18
(±0.19) |
| **GNRF**$_{FRC}$ | 85.59
(±1.56) | 80.00
(±10.50) | 82.08
(±5.41) | 75.23
(±0.68) | 76.17
(±0.46) | 81.61
(±1.07) | 61.78
(±0.99) | 41.22
(±0.43) | 67.51
(±0.87) | 88.96
(±0.14) | 82.55
(±0.32) | 87.29
(±0.55) |
| **GNRF**$_{ARC}$ | 86.49
(±2.70) | 88.00
(±2.00) | 81.90
(±5.63) | 76.52
(±0.33) | 78.14
(±0.32) | 87.25
(±1.01) | 64.55
(±1.33) | 41.74
(±0.46) | 70.17
(±0.61) | 88.21
(±0.40) | 83.83
(±0.45) | 89.43
(±0.22) |

Table 1: We compare GNRF with two classes of methods on the node classification task. Highlighted are the top first, second, and third results. OOM means out of memory. Accuracy is the measure for the vast majority of datasets, and for Minesweeper, Tolokers, and Questions, we use ROC-AUC.

**Main results (Table 1).** GNRF consistently demonstrates outstanding performance across diverse datasets. Compared to Cora_Full, PubMed and Cora_ML, GNRF shows significant improvements of over 20% compared to GCN on the other 8 heterophilic datasets. This improvement is attributed to GNRF forcing positive curvature edges to repel each other, as ACMP also performs well with a similar repulsive bias. However, since ACMP by default uses an adaptive step solver (Dormand-Prince 5), it excessively subdivides the step size when solving near-stiff equations, leading to an OOM. In contrast, GNRF remains stable on these datasets. Another observation is that GNRF$_F RC$ and GNRF$_A RC$ indeed show significant improvements over classic algorithms (such as GCN, GAT, GRAND, GRAND++), but they still fall short compared to GNRF using EdgeNet. This suggests that Attri-DRF itself is beneficial for graph learning, but adaptive curvature can maximize its utility.

**Ablation study (Table 2).** Two key modules: EdgeNet (e) and the damping factor (d), are used for ablation. When we remove EdgeNet from GNRF, we replace the aggregation weights with a trainable positive scalar. The results show that EdgeNet provides the primary improvement of GNRF, while the damping factor ensures numerical stability. Due to the inability to utilize negative weights, GRAND performs poorly on two heterogeneous datasets—Roman-Empire and Tolokers—and adjusting the damping factor does not lead to additional performance improvement. In contrast, ACMP performs much better, as it is similar to GRAND but can leverage negative weights. GNRF further improves upon ACMP, as it enables more fine-grained control over the weight signs—achieved through Ricci flow. On the other hand, ACMP uses an adaptive step-size solver, making it difficult to effectively handle near-stiff equations. However, when the damping factor is introduced, it produces stable and competitive numerical results.

**Resource consumption (Table 3).** We validated the scalability of GNRF on the OGBN-Arxiv dataset. We fixed the depth of all compared methods to 3 and reported the model parameter count, runtime per epoch (averaged over 1000 iterations), and accuracy for hidden layer sizes of 16, 64, and 256. For GCN, we used the hyperparameters recommended by the official OGB guidelines (Hu et al., 2020); for GAT, we used the same hyperparameters as GCN and set the number of attention heads to 3. We performed full-batch training on OGBN-Arxiv. Our results show that, for the same model capacity (i.e., the same hidden layer size), GNRF outperforms the other three methods. The

| ODE Solver | Dormand-Prince 5 (adaptive step) | | Adams-Moulton (fix step) | | | | | | |
|---|---|---|---|---|---|---|---|---|---|
| Method | ACMP | ACMP+d | ACMP | ACMP+d | GNRF | GNRF/d | GNRF/e | GRAND | GRAND+d |
| Roman-Empire | OOM | OOM | NC | 72.44 | 86.25 | NC | 53.71 | 60.12 | 58.57 |
| Tolokers | OOM | OOM | NC | 80.33 | 83.96 | NC | 78.60 | 79.01 | 78.78 |
| Cora_Full | OOM | OOM | NC | 71.17 | 72.12 | NC | 71.55 | 67.66 | 67.31 |

Table 2: NC means Not Convergent. "d" and "e" represent the damping factor and EdgeNet respectively. We use "+" or "/" to denote adding or removing a module.

| | | #hidden=16 | | | #Hidden=64 | | | #Hidden=256 | | |
|---|---|---|---|---|---|---|---|---|---|---|
| | | #Param | Time | Acc. | #Param | Time | Acc. | #Param | Time | Acc. |
| Depth=3 | GCN | 3.15k | 0.12s | 60.95 | 15.2k | 0.14s | 68.55 | 110k | 0.21s | **71.65** |
| | GAT | 15.0k | 0.17s | 59.52 | 86.8k | 0.25s | 64.39 | 788k | OOM | OOM |
| | ACMP | 4.18k | 3.29s | 61.03 | 32.0k | 6.35s | 68.89 | 374k | OOM | OOM |
| | GNRF | 5.50k | 0.31s | **62.11** | 52.6k | 0.78s | **69.33** | 701k | OOM | OOM |

Table 3: With a fixed depth of 3, we compare the performance and scalability of GNRF with other models at hidden layer sizes of 16, 64 and 256 respectively.

number of learnable parameters in GNRF lies between those of GCN and GAT, and its training time is comparable to theirs, significantly faster than ACMP (the current state-of-the-art continuous deep GNN model). At a hidden layer size of 256, GAT, ACMP, and GNRF all experienced out-of-memory. We conclude that GNRF strikes a meaningful trade-off between the high scalability but low accuracy of classical discrete deep GNNs and the low scalability but high accuracy of continuous-depth GNNs.

## 5.2 CURVATURE

We demonstrate whether GNRF faithfully adheres to the theoretical guidance provided in Chapter 3. We measure the curvature distribution of all edges in the graph dataset at different time points (Figure 2 above) and record the variance of these curvatures over time (Figure 2 below). We perform a 0-1 normalization of the variance from $t = 0$ to $t = 20$ to scale multiple datasets to a unified scale.

As the results show, in homophilic graphs (Cora-Full and Pubmed), positive curvature edges dominate, while in heterophilic graphs (Roman-empire and Tolokers), negative curvature holds more weight. The reason is that in homophilic graphs, nodes of the same class tend to form tightly connected community structures, which lead to edges with positive curvature (Topping et al., 2021).

However, regardless of the curvature distribution, as predicted by Lemma 1, the GNRF designed based on Attir-DRF will force them to concentrate around zero. This avoids the over-smoothing or information bottleneck issues caused by extreme curvature (Nguyen et al., 2023). Another observation is that the speed of this "concentration towards zero" is roughly the same across different datasets, as their curvature variances follow a similar inverse relationship, achieving relatively stable values around $t = 10$. This means that regardless of the dataset, the GNRF can always

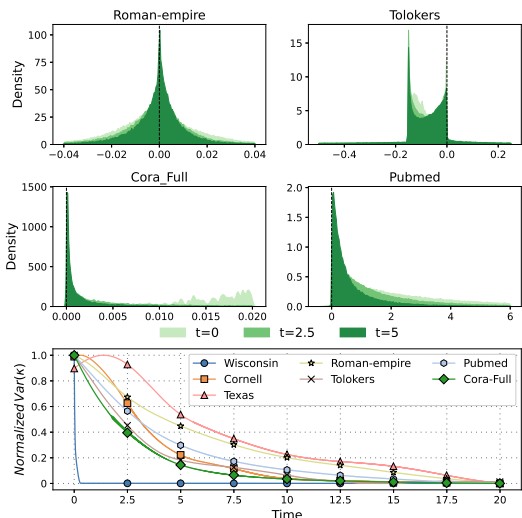

Figure 2: The distribution of curvature across different datasets (above) and the variation of their variance over time (below).

achieve stable curvature within a feasible time, indicating that the model designed based on Attri-DRF is consistently efficient in the evolution of node features and aligns with the expectations of Theorem 3.

## 5.3 DIRICHLET ENERGY AND FEATURE VISUALIZATION

We present the evolution of the Dirichlet energy of GNRF with random parameters on synthetic graphs without any training. The synthetic graph consists of 10,000 nodes, where any pair of nodes has a connection probability of 0.5, and the node features are sampled from a 10-dimensional standard Gaussian distribution. We measure the energy variations of GNRF compared to GCN (Kipf & Welling, 2016), GAT (Veličković et al., 2017), and GRAND (Chamberlain et al., 2021) in the random graph. Specifically, to understand the impact of the damping factor (DF), two derived models—GNRF without DF and GRAND with DF—are also used for comparison (Figure 3).

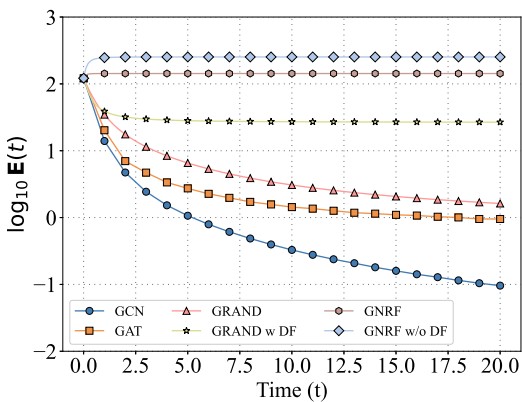

The features of GCN, GAT, and GRAND become significantly smoother with the increase of time (for GCN and GAT, time corresponds to the number of layers). However, the damping factor can reduce the length of the update gradient when features become similar, allowing GRAND to achieve a non-zero energy lower bound. For GNRF, even without the DF, it will lead to a decrease in $\|\partial_t \boldsymbol{h}_i(t)\|$ as $\|\text{EdgeNet}_{ij}(t)\|$ approaches 0 (due to curvature approaches 0), which similarly causes the Dirichlet energy to converge to a positive value. Under the combined influence of the damping factor and curvature, the Dirichlet energy of GNRF exhibits near-steady-state behavior, which is also within the scenarios envisioned in Theorem 2.

Figure 3: The trend of Dirichlet energy changes over time across different models.

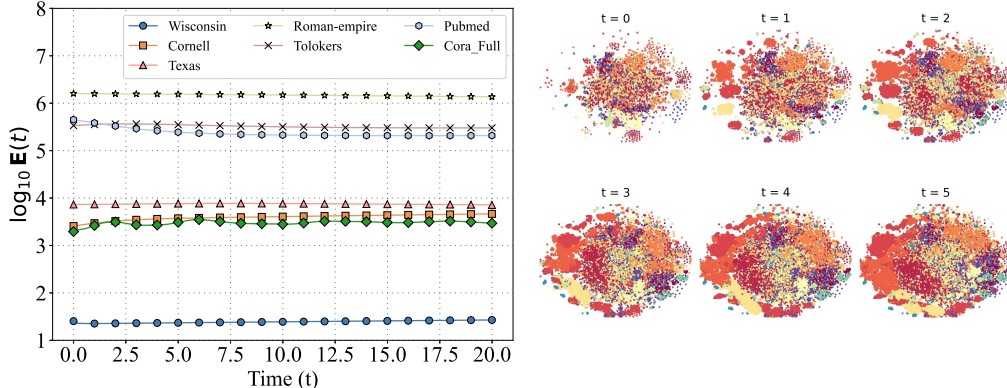

Figure 4: The evolution of Dirichlet energy of GNRF across different datasets (left); visualization of node representations for Roman-Empire (right).

Next, we show the Dirichlet energy evolution of the well-trained GNRF under each dataset (Fig. 4, left). The results illustrate that the GNRF guarantees that the Dirichlet energy is nearly constant, regardless of whether it is trained. However, constant energy does not mean stagnation of feature evolution. We use t-SNE to visualize the node features of the Roman-empire dataset (Fig. 4 right) and found that nodes of the same category do form larger and larger clusters, suggesting that the GNRF is beneficial for classification. We select two classes of node features in the dataset for visualization (Fig. 5) to further illustrate the mechanism by which GNRF takes effect. When we look at the data at $t = 0$, we see that nodes of the same class form many very tight clusters. Moreover, the clus-

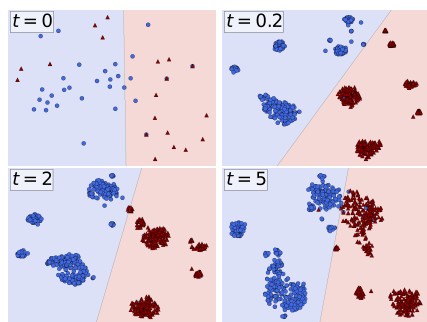

Figure 5: Visualization of node features for Roman-Empire's Class 11 and 13.

ters of the same class are closer to each other than the clusters of different classes. As time evolves, small tight clusters gradually become loose and merge with other clusters of the same class to form larger wholes. This, in turn, leads to more regular decision boundaries. However, we also recognize that too long an evolutionary time can exacerbate category indistinguishability and, therefore, requires careful tradeoffs.

## 6    CONCLUSION

We propose a novel continuous-depth graph neural network called Graph Neural Ricci Flow (GNRF). Specifically, we find that if the classical discrete Ricci flow (DRF) is generalized to attributed graphs, this benefits the evolution of node features, including Dirichlet energy with bilateral bounds and data-independent evolution time. GNRF is built on the attribute DRF and is the first graph deep learning model to use time-varying edge curvature. GNRF can fit edge curvature in linear time and adjust it to near zero, which prevents over-smoothing and information bottlenecks.

ACKNOWLEDGMENTS

The National Natural Science Foundation of China under Grant 62176269 and The National Natural Science Foundation of China under Grant 62302537 support the research.

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

# A RELATED WORK

## A.1 RIEMANNIAN GRAPH LEARNING

Recent research has shown that studying GNNs from the perspective of Riemannian geometry can provide additional benefits. For instance, embedding graph data into a hyperbolic space can effectively address the neighborhood explosion problem that arises when performing message aggregation on graphs with a power-law distribution (Yang et al., 2022; Sun et al., 2021; Chami et al., 2019). The core idea of such work is to map representations into a space of constant negative curvature (in contrast to the zero curvature of Euclidean space), thereby improving representation quality. Another more direct approach is to learn based on general Riemannian manifolds. They use graph curvature as a measurement for graph topology, and achieve structure-aware learning through graph rewiring (Nguyen et al., 2023; Topping et al., 2021; Fesser & Weber, 2024), edge sampling (Liu et al., 2023), or neighbor reweighting (Li et al., 2022).

## A.2 EDGE CURVATURE ON WEIGHTED GRAPHS

**Forman-Ricci curvature.** The idea of transferring the concept of curvature from Riemannian geometry to discrete graph-structured data was initially proposed by Forman (2003). This paper suggests constructing an analog of curvature in graph spaces using $k$-dimensional discrete CW complexes and their weights. A commonly used definition considers 1-dimensional and 2-dimensional complexes (Sreejith et al., 2016), which is:

$$\kappa_{ij}^{\text{FR}} = w_i + w_j - w_{ij} \sum_{u \sim i} \sqrt{\frac{w_i}{w_{iu}}} - w_{ij} \sum_{v \sim j} \sqrt{\frac{w_j}{w_{jv}}}. \tag{11}$$

Here, $w_i$ represents the weight on node $i$. However, in many cases, we only have edge weights without node weights. A simple approach in this situation is to consider the node weight as the sum of the weights of all its connected edges:

$$\kappa_{ij}^{\text{FR}} = 2 - \sum_{u \sim i} \frac{w_{ij}}{\sqrt{w_{iu}}} - \sum_{v \sim j} \frac{w_{ij}}{\sqrt{w_{jv}}}. \tag{12}$$

Although there have been several improvements to Forman-Ricci curvature, the most notable ones being balanced Forman curvature (Topping et al., 2021) and augmented Forman curvature (Iváñez, 2022) , they cannot be defined on weighted graphs.

**Resistance curvature.** Resistance curvature is defined based on the equivalent resistance in a resistance network. We envision a network composed of multiple resistors connected, where each edge in the network has an associated resistor, and the positive edge weights represent the resistance values. Suppose we measure the voltage and current between any two vertices externally. In that case, the network can be simplified to a single equivalent resistor, and the resistance value of this resistor is referred to as the equivalent resistance. Equivalent resistance is a form of distance (Devriendt & Lambiotte, 2022; Devriendt et al., 2024), and compared to other distance metrics defined on edge-weighted graphs, it effectively captures the connectivity information of the entire network. Specifically, if we define the following weighted Laplacian matrix $\boldsymbol{Q}$:

$$(\boldsymbol{Q})_{ij} = \begin{cases} -w_{ij} & \text{if} \quad i \sim j \\ \Sigma_{j \sim i} w_{ij} & \text{if} \quad i = j \\ 0 & \text{otherwise.} \end{cases} \tag{13}$$

then the equivalent resistance between nodes $i$ and $j$ can be defined as:

$$r_{ij} = (\boldsymbol{e}_i - \boldsymbol{e}_j)^T \boldsymbol{Q}^+ (\boldsymbol{e}_i - \boldsymbol{e}_j). \tag{14}$$

where $\boldsymbol{e}_i$ represents the $i$-th unit vector, and $Q^+$ denotes the pseudo-inverse of $Q$. Resistance distance helps understand the robustness of GNNs and addresses the overs-quashing problem (Shen et al., 2024; Black et al., 2023); it can also be used to define new discrete curvatures (Devriendt & Lambiotte, 2022; Devriendt et al., 2024). In this paper, we use the definition proposed by Devriendt & Lambiotte (2022), which is:

$$\kappa_{ij}^{\text{RC}} = \frac{2 - \sum_{k \sim i} r_{ik} w_{ik} - \sum_{k \sim j} r_{ik} w_{jk}}{r_{ij}}. \tag{15}$$

Although this definition seems appealing, it is not computable in real-time. The reason is that calculating the pseudo-inverse of a matrix requires at least cubic time complexity. Fortunately, Radl et al. (2009) provides us with a simple approximation:

$$r_{ij} \approx \widetilde{r}_{ij} = \frac{1}{N}\left(\frac{1}{\sum_{u\sim i} w_{ui}} + \frac{1}{\sum_{v\sim j} w_{vj}}\right). \tag{16}$$

Where $N$ represents the number of nodes in the graph. This formula guarantees an error of $\mathcal{O}(1/N)$. The curvature obtained using this approximation is referred to as the **approximate resistance curvature**:

$$\widetilde{\kappa}_{ij}^{\mathrm{RC}} = \frac{2 - \sum_{k\sim i} \widetilde{r}_{ik} w_{ik} - \sum_{k\sim j} \widetilde{r}_{ik} w_{jk}}{\widetilde{r}_{ij}}. \tag{17}$$

**Ollivier-Ricci curvature.** Since Ollivier (2007) introduced this concept, ORC has become one of the most commonly used mathematical tools for analyzing networks using geometric methods. It has been widely applied in various areas, including complex network analysis (Paeng, 2012), community detection (Sia et al., 2019), hypergraph learning (Coupette et al., 2022), and understanding over-smoothing and over-squashing (Nguyen et al., 2023). A more accessible definition can be found in Hehl (2024). Specifically, let us define the probability measure of a random walk originating from any node $u$ within its first-order neighbors as:

$$\mu_u(v) = \begin{cases} \dfrac{w_{xy}}{\sum_{z\sim x} w_{xz}} & \text{if} \quad y \sim x \\ 0, & \text{otherwise.} \end{cases} \tag{18}$$

Then, the $\kappa_{ij}^{\mathrm{OR}}$ between any node pair $(i, j)$ is defined as:

$$\kappa_{ij}^{\mathrm{OR}} = 1 - \frac{W_1(\mu_i, \mu_j)}{d(i, j)}. \tag{19}$$

where $d(i, j)$ represents the shortest distance between nodes $i$ and $j$ in graph $\mathcal{G}$. For adjacent nodes $i$ and $j$, $d(i, j) = 1$. $W_1$ denotes the 1-Wasserstein distance between the two probability measures.

## A.3 DIRICHLET ENERGY

Dirichlet energy is a commonly used measure of the smoothness of node attributes (Chen et al., 2020a; Zhao & Akoglu, 2019), and it is also employed as a regularization term to mitigate GNN over-smoothing (Zhou et al., 2021b). First, we define the symmetrically normalized adjacency matrix for graph $\mathcal{G}$ as follows:

$$\widetilde{\boldsymbol{A}} = \begin{cases} \dfrac{1}{\sqrt{(1+d_i)(1+d_j)}}, & i \sim j, \\ 0, & \text{otherwise.} \end{cases} \tag{20}$$

Here, $d_i$ denotes the degree of node $i$. When the node attribute matrix on $\mathcal{G}$ is $\boldsymbol{H}$, the Dirichlet energy is defined as:

$$E(\boldsymbol{H}) = \mathrm{Tr}\big(H^T(\boldsymbol{I} - \widetilde{\boldsymbol{A}})H\big) = \frac{1}{2} \sum_{(i,j)\in\mathcal{E}} \left\| \frac{\boldsymbol{h}_i}{\sqrt{1+d_i}} - \frac{\boldsymbol{h}_j}{\sqrt{1+d_j}} \right\|_2^2 \tag{21}$$

## B PROOF

**Lemma B.1.** *If there exists a constant $L$ such that for any $x_1, x_2 \in [x_1, x_\mathrm{r}]$, the continuous function $f$ satisfies $|f(x_1) - f(x_2)| \geq L|x_1 - x_2|$, then $f(x)$ is monotonic in $[x_1, x_\mathrm{r}]$.*

*Proof.* Consider a proof by contradiction. Suppose $f$ is not monotonic. Then a critical point $x^* \in (x_1, x_\mathrm{r})$ must exist. Without loss of generality, assume $f(x^*)$ is a local maximum. Then there must exist points $x_-^*$ and $x_+^*$ on either side of $x^*$ such that $f(x^*) > f(x_-^*) = f(x_+^*)$.

$$0 = |f(x_+^*) - f(x_-^*)| \geq L|x_+^* - x_-^*| > 0. \tag{22}$$

This leads to a contradiction, so $f(x)$ cannot have any critical points in $[x_1, x_\mathrm{r}]$, which implies that $f(x)$ is monotonic. $\qquad\square$

**Lemma B.2.** *(Maruhashi et al., 2012) Let $\boldsymbol{A}$ denote the unweighted adjacency matrix of an undirected graph $\mathcal{G}$, $\boldsymbol{v}_r$ the $r$-th eigenvector of $\boldsymbol{A}$, $\boldsymbol{\lambda}$ the vector composed of all eigenvalues of $\boldsymbol{A}$. $\boldsymbol{\lambda}^k$ implies that the $k$-th power is applied to each element of $\boldsymbol{\lambda}$. Then the shortest path distance between any two nodes $i$ and $j$ is:*

$$d(i,j) = \min_k \left[ \left( \boldsymbol{v}_i \circ \boldsymbol{\lambda}^k \circ \boldsymbol{v}_j \right)^T \boldsymbol{1} > 0 \right]. \tag{23}$$

**Lemma B.3.** *(Chen & Wang, 2024) Let $\epsilon > 0$, $(\Omega, d_\Omega)$ be a compact metric space and let $\mathcal{X}$ be the space of weighted point sets equipped with $p$-Wasserstein, for any $A, B \in \mathcal{X}$, there exist trainable networks $\phi_1$, $\phi_2$, and $\phi_3$ with sufficiently large numbers of parameters such that:*

$$\left| W_p(A,B) - \phi_1 \left( \phi_2 \left( \sum_{(x,w_x) \in A} w_x \phi_3(x) \right) + \phi_2 \left( \sum_{(y,w_y) \in B} w_y \phi_3(y) \right) \right) \right| < \epsilon. \tag{24}$$

**Lemma B.4.** *For any symmetric adjacency matrix $\boldsymbol{A} \in \{0,1\}^{n \times n}$, and its corresponding degree matrix $\boldsymbol{D} = \mathrm{diag}(d_1, \cdots, d_n)$, the following inequality holds:*

$$\sum_{i \in [n]} \frac{d_i}{1 + d_i} - \sum_{i \in [n]} \sum_{j \in [n]} \frac{\boldsymbol{A}_{ij}}{\sqrt{(1+d_i)(1+d_j)}} \geq 0 \tag{25}$$

*The equality sign is taken when and only when $\boldsymbol{A}$ is a regular graph.*

*Proof.* Notes that:

$$\sum_{i \in [n]} \frac{d_i}{1 + d_i} = \sum_{i \in [n]} \frac{d_i}{\sqrt{(1+d_i)(1+d_i)}} = \boldsymbol{1}_n^T (\boldsymbol{I} + \boldsymbol{D})^{-\frac{1}{2}} \boldsymbol{D} (\boldsymbol{I} + \boldsymbol{D})^{-\frac{1}{2}} \boldsymbol{1}_n, \tag{26}$$

and

$$\sum_{i \in [n]} \sum_{j \in [n]} \frac{\boldsymbol{A}_{ij}}{\sqrt{(1+d_i)(1+d_j)}} = \boldsymbol{1}_n^T (\boldsymbol{I} + \boldsymbol{D})^{-\frac{1}{2}} \boldsymbol{A} (\boldsymbol{I} + \boldsymbol{D})^{-\frac{1}{2}} \boldsymbol{1}_n. \tag{27}$$

Thus the inequality to be proved can be transformed into:

$$\boldsymbol{1}_n^T (\boldsymbol{I} + \boldsymbol{D})^{-\frac{1}{2}} (\boldsymbol{D} - \boldsymbol{A}) (\boldsymbol{I} + \boldsymbol{D})^{-\frac{1}{2}} \boldsymbol{1}_n \geq 0. \tag{28}$$

Let $\boldsymbol{x} = 1_n (\boldsymbol{I} + \boldsymbol{D})^{-\frac{1}{2}}$ and $\boldsymbol{L} = \boldsymbol{D} - \boldsymbol{A}$, now we need to prove:

$$\boldsymbol{x}^T \boldsymbol{L} \boldsymbol{x} \geq 0. \tag{29}$$

This is clearly valid according to the semi-positive characterization of the Laplace matrix.

We now consider the conditions under which the equal sign holds. From linear algebra, we know that the quadratic form $\boldsymbol{x}^T \boldsymbol{L} \boldsymbol{x}$ is equal to 0 if and only if $\boldsymbol{x}$ lies in the null space of $\boldsymbol{L}$, i.e., $\boldsymbol{L}\boldsymbol{x} = \boldsymbol{0}$. More specifically, $(\boldsymbol{D} - \boldsymbol{A})(\boldsymbol{I} + \boldsymbol{D})^{-\frac{1}{2}} \boldsymbol{1}_n = 0$. Expanding it into elemental form, we get that for every node $i$, it must satisfy:

$$\frac{d_i}{\sqrt{1+d_i}} = \sum_{j \sim i} \frac{1}{\sqrt{1+d_j}}. \tag{30}$$

Obviously, when $\boldsymbol{A}$ is a regular graph, i.e., $d_i \equiv r$, this condition always holds. However, if $\boldsymbol{A}$ is not a regular graph, then there must exist a node in $\boldsymbol{A}$ with the highest degree among all nodes, and it is connected to at least one node with a lower degree. Let us denote such a node as $i$. In this case, we can conclude:

$$\frac{d_i}{\sqrt{1+d_i}} = \sum_{j \sim i} \frac{1}{\sqrt{1+d_j}} > \sum_{j \sim i} \frac{1}{\sqrt{1+d_i}} = \frac{d_i}{\sqrt{1+d_i}}. \tag{31}$$

This leads to a contradiction, meaning that the equality can only hold when $\boldsymbol{A}$ is a regular graph. $\quad\square$

### B.1 PROOF OF LEMMA 1

**Lemma B.5** (Formal version of lemma 1). *Consider the Attri-DRF over the interval $[t_1, t_2]$. If the edge weight $w_{ij}(t)$ has a finite number of $N$ extrema points within $(t_1, t_2)$, denoted by $T_1, \ldots, T_N$, and define $T_0 = t_1$ and $T_{N+1} = t_2$, then:*

$$\frac{\lambda - 1}{\zeta} > \mathbb{E}_{t \in [t_1, t_2]}(|\kappa_{ij}(t)|) > \frac{1 - \lambda^{-1}}{t_2 - t_1}, \tag{32}$$

*where $\zeta = \min_{0 \leq i \leq N}(T_{i+1} - T_i)$ represents the length of the most minor maximal monotonic interval within $[t_1, t_2]$, and $\lambda = \frac{\max_a w(T_a)}{\min_b w(T_b)}$ denotes the ratio of the maximum to the minimum value of $w(t)$. Specifically, if $w_{ij}(t)$ is monotonic within $(t_1, t_2)$, we have:*

$$\frac{1}{\min\{w(t_2), w(t_1)\}} \frac{|w(t_2) - t(t_1)|}{t_2 - t_1} > \mathbb{E}_{t \in [t_1, t_2]}(|\kappa_{ij}(t)|), \tag{33}$$

*which $|w(t_2) - t(t_1)| \to 0$ implies $\mathbb{E}_{t \in [t_1, t_2]}(|\kappa_{ij}(t)|) \to 0$.*

Since the edge $i \sim j$ is arbitrary, we omit the $\kappa_{ij}(t)$ subscript in the proof. First, we integrate Equation (3) over an arbitrary time interval $[t_1, t_2]$:

$$w(t_2) - w(t_1) = -\int_{t_1}^{t_2} \kappa(t) w(t) dt. \tag{34}$$

Suppose that within $[t_1, t_2]$, the function $w(t)$ has $N$ extremum points. These $N$ points divide the interval $[t_1, t_2]$ into $N + 1$ adjacent monotonic sub-intervals. Given that $w(t) > 0$, according to Equation (1), it follows that the sign of $\frac{\partial w(t)}{\partial t}$ is determined by $\kappa(t)$. Moreover, the $\kappa(t)$ sign remains non-negative or non-positive within each monotonic sub-interval.

Let $[T_i, T_{i+1}]$ denote any one of these $N + 1$ monotonic sub-intervals. If $\kappa(t) \geq 0$ for $t \in [T_i, T_{i+1}]$, then $w(t)$ is monotonically decreasing, which implies $w(T_i) > w(t) > w(T_{i+1}) > 0$. Therefore:

$$0 > -w(T_{i+1}) \int_{T_i}^{T_{i+1}} \kappa(t) dt > -\int_{T_i}^{T_{i+1}} \kappa(t) w(t) dt > -w(T_i) \int_{T_i}^{T_{i+1}} \kappa(t) dt. \tag{35}$$

If $\kappa(t) \leq 0$ for $t \in [T_i, T_{i+1}]$, then $w(t)$ is monotonically increasing, which implies $w(T_{i+1}) > w(t) > w(T_i) > 0$ and

$$-w(T_{i+1}) \int_{T_i}^{T_{i+1}} \kappa(t) dt > -\int_{T_i}^{T_{i+1}} \kappa(t) w(t) dt > -w(T_i) \int_{T_i}^{T_{i+1}} \kappa(t) dt > 0. \tag{36}$$

Therefore, on any monotonic sub-interval, we have:

$$-w(T_{i+1}) \int_{T_i}^{T_{i+1}} \kappa(t) dt > w(T_{i+1}) - w(T_i) > -w(T_i) \int_{T_i}^{T_{i+1}} \kappa(t) dt. \tag{37}$$

Take the absolute value of the above inequality:

$$\max\{w(T_i), w(T_{i+1})\} \int_{T_i}^{T_{i+1}} |\kappa(t)| dt > |w(T_{i+1}) - w(T_i)| > \min\{w(T_i), w(T_{i+1})\} \int_{T_i}^{T_{i+1}} |\kappa(t)| dt. \tag{38}$$

Let $w_{\max} > \max\{w(T_i), w(T_{i+1})\} > \min\{w(T_i), w(T_{i+1})\} > w_{\min}$, we have:

$$\frac{|w(T_{i+1}) - w(T_i)|}{w_{\min}} > \int_{T_i}^{T_{i+1}} |\kappa(t)| dt > \frac{|w(T_{i+1}) - w(T_i)|}{w_{\max}}. \tag{39}$$

Summing over $i$ from 0 to $N$ and then dividing by $t_2 - t_1$:

$$\frac{\sum_{i=0}^{N} |w(T_{i+1}) - w(T_i)|}{w_{\min}(t_2 - t_1)} > \frac{1}{t_2 - t_1} \int_{t_1}^{t_2} |\kappa(t)| dt = \mathbb{E}_{t \in [t_1, t_2]}(|\kappa_{ij}(t)|) > \frac{\sum_{i=0}^{N} |w(T_{i+1}) - w(T_i)|}{w_{\max}(t_2 - t_1)}. \tag{40}$$

Because $(N + 1)(w_{\max} - w_{\min}) > \sum_{i=0}^{N} |w(T_{i+1}) - w(T_i)| > w_{\max} - w_{\min}$, and $\lambda = \frac{w_{\max}}{w_{\min}}$, then:

$$\frac{(N + 1)(\lambda - 1)}{t_2 - t_1} > \mathbb{E}_{t \in [t_1, t_2]}(|\kappa_{ij}(t)|) > \frac{1 - \lambda^{-1}}{t_2 - t_1}. \tag{41}$$

According to the definition of $\zeta$, we have $(t_2 - t_1) > \zeta(N + 1)$. Thus, the estimation of the upper bound can be replaced by $\zeta^{-1}(\lambda - 1) > \frac{(N+1)(\lambda-1)}{t_2-t_1}$, thereby completing the proof.

## B.2 PROOF OF THEOREM 2

**Theorem B.6.** *Consider the Attri-DRF with edge weight $w_{ij}(t) \equiv \cos(\boldsymbol{h}_i, \boldsymbol{h}_j) + 1 + \epsilon$ and $|\boldsymbol{h}(t)| \equiv 1$. If for any $i \sim j$ in $\mathcal{G}$, $w_{ij}(t)$ has a finite number of monotonic intervals on $[t_1, t_2]$, then the average Dirichlet energy within $[t_1, t_2]$ has following bound:*

$$B_1 \geq \mathbb{E}_{t \in [t_1, t_2]}(E(\boldsymbol{H}(t))) \geq B_2, \tag{42}$$

*where:*

$$
\begin{aligned}
B_1 &= (t_2 - t_1) \sum_{i \in \mathcal{V}} \frac{\sum_{j \sim i} \lambda_{ij} \zeta_{ij}^{-1}}{1 + d_i} + \sum_{(i,j) \in \mathcal{E}} ((t_2 - t_1)\lambda_{ij}\zeta_{ij}^{-1}(1 + \epsilon) - \epsilon\lambda_{ij})\widetilde{\boldsymbol{A}}_{ij}, \\
B_2 &= \sum_{(i,j) \in \mathcal{E}} \widetilde{\boldsymbol{A}}_{ij} \left(1 + \epsilon - (2 + \epsilon)\zeta_{ij}^{-1}(t_2 - t_1)\right) + \sum_{i \in \mathcal{V}} \frac{d_i}{1 + d_i}.
\end{aligned}
\tag{43}
$$

*$d_i$ denotes the degree of node $i$, and $\widetilde{\boldsymbol{A}}$ represents the symmetrically normalized adjacency matrix of graph $\mathcal{G}$.*

We begin the proof by expanding the definition of the Dirichlet energy. Noting that $\|\boldsymbol{h}_i(t)\| = 1$ and $w_{ij}(t) = \boldsymbol{h}_i(t)^T \boldsymbol{h}_j(t) + 1 + \epsilon$, we have:

$$
\begin{aligned}
\boldsymbol{E}(\boldsymbol{H}(t)) &= \frac{1}{2} \sum_{(i,j) \in \mathcal{E}} \left\| \frac{\boldsymbol{h}_i(t)}{\sqrt{1 + d_i}} - \frac{\boldsymbol{h}_j(t)}{\sqrt{1 + d_j}} \right\|_2^2 \\
&= \frac{1}{2} \sum_{(i,j) \in \mathcal{E}} \left( \frac{\|\boldsymbol{h}_i(t)\|^2}{1 + d_i} + \frac{\|\boldsymbol{h}_j(t)\|^2}{1 + d_j} - 2 \frac{\boldsymbol{h}_i(t)^T \boldsymbol{h}_j(t)}{\sqrt{(1 + d_i)(1 + d_j)}} \right) \\
&= \sum_{(i,j) \in \mathcal{E}} \left[ \frac{1}{2} \left( \frac{1}{1 + d_i} + \frac{1}{1 + d_j} \right) + \frac{1 + \epsilon}{\sqrt{(1 + d_i)(1 + d_j)}} - \frac{w_{ij}(t)}{\sqrt{(1 + d_i)(1 + d_j)}} \right].
\end{aligned}
\tag{44}
$$

By defining $c_{ij} \triangleq \frac{1}{2}\left(\frac{1}{1+d_i} + \frac{1}{1+d_j}\right) + \frac{1+\epsilon}{\sqrt{(1+d_i)(1+d_j)}}$ and $\boldsymbol{E}_{ij}(t) \triangleq c_{ij} - \frac{w_{ij}(t)}{\sqrt{(1+d_i)(1+d_j)}}$, the Dirichlet energy can also be simplified to:

$$\boldsymbol{E}(\boldsymbol{H}(t)) = \sum_{(i,j) \in \mathcal{E}} \left( c_{ij} - \frac{w_{ij}(t)}{\sqrt{(1 + d_i)(1 + d_j)}} \right) = \sum_{(i,j) \in \mathcal{E}} \boldsymbol{E}_{ij}(t). \tag{45}$$

Take the partial derivative of the above equality to $t$:

$$
\begin{aligned}
\frac{\partial \boldsymbol{E}(\boldsymbol{H}(t))}{\partial t} &= \sum_{(i,j) \in \mathcal{E}} \left( -\frac{1}{\sqrt{(1 + d_i)(1 + d_j)}} \frac{\partial w_{ij}(t)}{\partial t} \right) \\
&= \sum_{(i,j) \in \mathcal{E}} \frac{\kappa_{ij}(t) w_{ij}(t)}{\sqrt{(1 + d_i)(1 + d_j)}} \\
&= \sum_{(i,j) \in \mathcal{E}} \kappa_{ij}(t) \left[ c_{ij} - \boldsymbol{E}_{ij}(t) \right].
\end{aligned}
\tag{46}
$$

Noting that the above equation holds for any graph $\mathcal{G}$, we can construct a subgraph $\mathcal{G}_{ij}$ such that it contains only two vertices $\{i, j\}$ and a single edge $i \sim j$. In this case, we have:

$$\frac{\partial \boldsymbol{E}_{ij}(t)}{\partial t} = \kappa_{ij}(t) \left[ c_{ij} - \boldsymbol{E}_{ij}(t) \right]. \tag{47}$$

Integrate over an arbitrary time interval $[t_1, t_2]$:

$$\boldsymbol{E}_{ij}(t_2) - \boldsymbol{E}_{ij}(t_1) = c_{ij} \int_{t_1}^{t_2} \kappa_{ij}(t) dt - \int_{t_1}^{t_2} \kappa_{ij}(t) \boldsymbol{E}_{ij}(t) dt. \tag{48}$$

We first prove the lower bound. Similar to the proof of Lemma 1, we consider the $N$ extremum points of $w_{ij}(t)$ within $(t_1, t_2)$: $T_1, \cdots, T_N$. For any $i$, $\kappa_{ij}(t)$ does not change its sign within $(T_i, T_{i+1})$, and since $\boldsymbol{E}_{ij}(t) \geq 0$, we have $\left| \int_{T_i}^{T_{i+1}} \kappa_{ij}(t)\boldsymbol{E}_{ij}(t)dt \right| = \int_{T_i}^{T_{i+1}} |\kappa_{ij}(t)|\boldsymbol{E}_{ij}(t)dt$, so:

$$
\begin{aligned}
|\boldsymbol{E}_{ij}(T_{i+1}) - \boldsymbol{E}_{ij}(T_i)| &= \left| c_{ij} \int_{T_i}^{T_{i+1}} \kappa_{ij}(t)dt - \int_{T_i}^{T_{i+1}} \kappa_{ij}(t)\boldsymbol{E}_{ij}(t)dt \right| \\
&= \left| c_{ij} \int_{T_i}^{T_{i+1}} \kappa_{ij}(t)dt \right| - \left| \int_{T_i}^{T_{i+1}} \kappa_{ij}(t)\boldsymbol{E}_{ij}(t)dt \right| \\
&= c_{ij} \int_{T_i}^{T_{i+1}} |\kappa_{ij}(t)|dt - \int_{T_i}^{T_{i+1}} |\kappa_{ij}(t)|\boldsymbol{E}_{ij}(t)dt \\
&\approx c_{ij} \int_{T_i}^{T_{i+1}} |\kappa_{ij}(t)|dt - \mathbb{E}_{t \in [t_1, t_2]}(|\kappa_{ij}(t)|) \int_{T_i}^{T_{i+1}} \boldsymbol{E}_{ij}(t)dt.
\end{aligned}
\tag{49}
$$

Sum over $i$ from 0 to $N$ and then divide by $t_2 - t_1$:

$$
\frac{1}{t_2 - t_1} \sum_{i=0}^{N} |\boldsymbol{E}_{ij}(T_{i+1}) - \boldsymbol{E}_{ij}(T_i)| \geq \mathbb{E}_{t \in [t_1, t_2]}(|\kappa_{ij}(t)|) \left[ c_{ij} - \mathbb{E}_{t \in [t_1, t_2]}(\boldsymbol{E}_{ij}(t)) \right].
\tag{50}
$$

Noting that $(N+1)\zeta_{ij} \leq t_2 - t_1$ and $w_{ij} \leq 2 + \epsilon$, the left-hand side of the above inequality can be enlarged to:

$$
\begin{aligned}
\frac{1}{t_2 - t_1} \sum_{i=0}^{N} |\boldsymbol{E}_{ij}(T_{i+1}) - \boldsymbol{E}_{ij}(T_i)| &= \frac{1}{t_2 - t_1} \sum_{i=0}^{N} \frac{|w_{ij}(T_{i+1}) - w_{ij}(T_i)|}{\sqrt{(1+d_i)(1+d_j)}} \\
&\leq \frac{(N+1)(\max w_{ij} - \min w_{ij})}{(t_2 - t_1)\sqrt{(1+d_i)(1+d_j)}} \\
&\leq \frac{\zeta_{ij}^{-1} \max w_{ij}(1 - \lambda_{ij}^{-1})}{\sqrt{(1+d_i)(1+d_j)}} \\
&\leq \frac{(2+\epsilon)\zeta_{ij}^{-1}(1 - \lambda_{ij}^{-1})}{\sqrt{(1+d_i)(1+d_j)}}.
\end{aligned}
\tag{51}
$$

Thus we have:

$$
\frac{(2+\epsilon)\zeta^{-1}(1 - \lambda^{-1})}{\sqrt{(1+d_i)(1+d_j)}} \geq \mathbb{E}_{t \in [t_1, t_2]}(|\kappa(t)|) \left[ c_{ij} - \mathbb{E}_{t \in [t_1, t_2]}(\boldsymbol{E}(t)) \right].
\tag{52}
$$

By applying Lemma B.1, we can obtain the lower bound for $\mathbb{E}_{t \in [t_1, t_2]}(\boldsymbol{E}_{ij}(t))$:

$$
\begin{aligned}
\mathbb{E}_{t \in [t_1, t_2]}(\boldsymbol{E}_{ij}(t)) &\geq c_{ij} - \frac{(2+\epsilon)\zeta_{ij}^{-1}(1 - \lambda_{ij}^{-1})}{\mathbb{E}_{t \in [t_1, t_2]}(|\kappa(t)|)\sqrt{(1+d_i)(1+d_j)}} \\
&= \left( 1 + \epsilon - \frac{(2+\epsilon)\zeta_{ij}^{-1}(1 - \lambda_{ij}^{-1})}{\mathbb{E}_{t \in [t_1, t_2]}(|\kappa_{ij}(t)|)} \right) \frac{1}{\sqrt{(1+d_i)(1+d_j)}} + \frac{1}{2}\left( \frac{1}{1+d_i} + \frac{1}{1+d_j} \right) \\
&= \widetilde{\boldsymbol{A}}_{ij} \left( 1 + \epsilon - \frac{(2+\epsilon)\zeta_{ij}^{-1}(1 - \lambda_{ij}^{-1})}{\mathbb{E}_{t \in [t_1, t_2]}(|\kappa_{ij}(t)|)} \right) + \frac{1}{2}\left( \frac{1}{1+d_i} + \frac{1}{1+d_j} \right) \\
&\geq \widetilde{\boldsymbol{A}}_{ij} \left( 1 + \epsilon - (2+\epsilon)\zeta_{ij}^{-1}(t_2 - t_1) \right) + \frac{1}{2}\left( \frac{1}{1+d_i} + \frac{1}{1+d_j} \right).
\end{aligned}
\tag{53}
$$

Take the sum over all edge:

$$
\mathbb{E}_{t \in [t_1, t_2]}(\boldsymbol{E}(t)) \geq \sum_{(i,j) \in \mathcal{E}} \widetilde{\boldsymbol{A}}_{ij} \left( 1 + \epsilon - (2+\epsilon)\zeta_{ij}^{-1}(t_2 - t_1) \right) + \sum_{i \in \mathcal{V}} \frac{d_i}{1+d_i}.
\tag{54}
$$

Now, let's derive the upper bound. Recalling Eq. (49), for any maximal monotonic interval $[T_i, T_{i+1}]$ of $w_{ij}(t)$, we have:

$$\int_{T_i}^{T_{i+1}} |\kappa_{ij}(t)| \boldsymbol{E}_{ij}(t) dt = c_{ij} \int_{T_i}^{T_{i+1}} |\kappa_{ij}(t)| dt - |\boldsymbol{E}_{ij}(T_{i+1}) - \boldsymbol{E}_{ij}(T_i)|. \tag{55}$$

Sum over $i$ from 0 to $N$:

$$\begin{aligned}
\int_{t_1}^{t_2} |\kappa_{ij}(t)| \boldsymbol{E}_{ij}(t) dt &= c_{ij} \int_{t_1}^{t_2} |\kappa_{ij}(t)| dt - \sum_{i=0}^{N} |\boldsymbol{E}_{ij}(T_{i+1}) - \boldsymbol{E}_{ij}(T_i)| \\
&= c_{ij} \int_{t_1}^{t_2} |\kappa_{ij}(t)| dt - \sum_{i=0}^{N} \frac{|w_{ij}(T_{i+1}) - w_{ij}(T_i)|}{\sqrt{(1+d_i)(1+d_j)}} \\
&\leq c_{ij} \int_{t_1}^{t_2} |\kappa_{ij}(t)| dt - \frac{\max w_{ij} - \min w_{ij}}{\sqrt{(1+d_i)(1+d_j)}} \\
&\leq c_{ij} \int_{t_1}^{t_2} |\kappa_{ij}(t)| dt - \frac{\epsilon(\lambda_{ij} - 1)}{\sqrt{(1+d_i)(1+d_j)}}.
\end{aligned} \tag{56}$$

Divide both sides by $t_2 - t_1$:

$$\mathbb{E}_{t \in [t_1, t_2]} \big( |\kappa_{ij}(t)| \boldsymbol{E}_{ij}(t) \big) \leq c_{ij} \mathbb{E}_{t \in [t_1, t_2]} \big( |\kappa_{ij}(t)| \big) - \frac{\epsilon(\lambda_{ij} - 1)}{(t_2 - t_1)\sqrt{(1+d_i)(1+d_j)}}. \tag{57}$$

Applying Lemma 1, we obtain:

$$\frac{1 - \lambda_{ij}^{-1}}{t_2 - t_1} \mathbb{E}_{t \in [t_1, t_2]} \big( \boldsymbol{E}_{ij}(t) \big) \leq c_{ij} \zeta_{ij}^{-1} (\lambda_{ij} - 1) - \frac{\epsilon(\lambda_{ij} - 1)}{(t_2 - t_1)\sqrt{(1+d_i)(1+d_j)}}. \tag{58}$$

Due to $\boldsymbol{E}(t) = \sum_{(i \sim j) \in \mathcal{E}} \boldsymbol{E}_{ij}(t)$, we can derive the upper bound as:

$$\begin{aligned}
\mathbb{E}_{t \in [t_1, t_2]} (\boldsymbol{E}(t)) &\leq (t_2 - t_1) \sum_{(i \sim j) \in \mathcal{E}} c_{ij} \lambda_{ij} \zeta_{ij}^{-1} - \sum_{(i \sim j) \in \mathcal{E}} \frac{\epsilon \lambda_{ij}}{\sqrt{(1+d_i)(1+d_j)}} \\
&= (t_2 - t_1) \left[ \sum_{(i,j) \in \mathcal{E}} \lambda_{ij} \zeta_{ij}^{-1} \left( \frac{1}{2}\left(\frac{1}{1+d_i} + \frac{1}{1+d_j}\right) + (1+\epsilon)\widetilde{\boldsymbol{A}}_{ij} \right) \right] - \epsilon \sum_{(i,j) \in \mathcal{E}} \lambda_{ij} \widetilde{\boldsymbol{A}}_{ij} \\
&= (t_2 - t_1) \sum_{i \in \mathcal{V}} \frac{\sum_{j \sim i} \lambda_{ij} \zeta_{ij}^{-1}}{1+d_i} + \sum_{(i,j) \in \mathcal{E}} ((t_2 - t_1) \lambda_{ij} \zeta_{ij}^{-1}(1+\epsilon) - \epsilon \lambda_{ij}) \widetilde{\boldsymbol{A}}_{ij}.
\end{aligned} \tag{59}$$

Combining the results of Eq. (53) and Eq. (59), we have:

$$\begin{aligned}
&(t_2 - t_1) \sum_{i \in \mathcal{V}} \frac{\sum_{j \sim i} \lambda_{ij} \zeta_{ij}^{-1}}{1+d_i} + \sum_{(i,j) \in \mathcal{E}} ((t_2 - t_1) \lambda_{ij} \zeta_{ij}^{-1}(1+\epsilon) - \epsilon \lambda_{ij}) \widetilde{\boldsymbol{A}}_{ij} \\
&\geq \mathbb{E}_{t \in [t_1, t_2]} (\boldsymbol{E}(t)) \geq \sum_{(i,j) \in \mathcal{E}} \widetilde{\boldsymbol{A}}_{ij} \left( 1 + \epsilon - (2+\epsilon)\zeta_{ij}^{-1}(t_2 - t_1) \right) + \sum_{i \in \mathcal{V}} \frac{d_i}{1+d_i}.
\end{aligned} \tag{60}$$

**Theorem B.7** (Formal version of Theorem 2). *Consider the Attri-DRF with edge weight $w_{ij}(t) \equiv \cos(\boldsymbol{h}_i, \boldsymbol{h}_j) + 1 + \epsilon$ and $|\boldsymbol{h}(t)| \equiv 1$. If $\mathcal{G}$ is a non-regular graph, and all $w_{ij}$ are monotonic on $[t_1, t_2]$, let $\lambda_{\max} = \max_{(i,j) \in \mathcal{E}} \lambda_{ij}$, we have:*

$$\lambda_{\max} \left( \sum_{i \in \mathcal{V}} \frac{d_i}{1+d_i} + \text{sum}(\widetilde{\boldsymbol{A}}) \right) \geq \mathbb{E}_{t \in [t_1, t_2]} (E(\boldsymbol{H}(t))) \geq \sum_{i \in \mathcal{V}} \frac{d_i}{1+d_i} - \text{sum}(\widetilde{\boldsymbol{A}}) > 0. \tag{61}$$

Note that if $\kappa(t)$ is monotonically approaching 0 within $[t_2, t_1]$, we have $t_2 - t_1 = \zeta$. Then this conclusion is a direct generalization of Theorem B.6 and Lemma B.4.

### B.3 PROOF OF THEOREM 3

> **Theorem B.8** (Formal version of Theorem 3). *Consider the Attri-DRF with $w_{ij}(t) \equiv \cos(\boldsymbol{h}_i, \boldsymbol{h}_j) + 1 + \epsilon$ and $|\boldsymbol{h}(t)| \equiv 1$ and $|\boldsymbol{h}| \equiv 1$ on any edge $i \sim j$. Assume that the curvature is bi-Lipschitz continuous when considered as a function of the weights, i.e., there exist $L$, $K$ such that $K|w(t_2) - w(t_1)| \geq |\kappa(t_2) - \kappa(t_1)| \geq L|w(t_2) - w(t_1)|$. Let $|\kappa_{ij}(0)| > 0$. For any arbitrarily small positive number $\delta$, it holds that:*
>
> $$\min_{t \in [0,+\infty)} \{|\kappa_{ij}(t)| = \delta\} \leq \frac{1}{L\epsilon - \delta} \ln\left(\frac{2L + \delta}{\delta(2 + \epsilon)}\right). \tag{62}$$

Similarly to the proof of Lemma 1, we omit the edge $i \sim j$ as a subscript. Since the proof process for $\kappa(0) < 0$ is entirely analogous to that for $\kappa(0) > 0$, we can assume $\kappa(0) > 0$ without loss of generality in the following proof.

Denote $\min_{t \in [0,+\infty)} \{|\kappa_{ij}(t)| = \delta\} = T^*$. Because $\kappa(t) > 0$ for all $t \in [0, T^*)$, then $\frac{\partial w(t)}{\partial t} < 0$ and $w(t) < w(0)$. In this case, the condition $|\kappa(t_2) - \kappa(t_1)| \geq L|w(t_2) - w(t_1)|$ will lead to the following two cases:

1. $\kappa(t) \geq \kappa(0) + L(w(0) - w(t))$
2. $\kappa(t) \leq \kappa(0) - L(w(0) - w(t))$

We first discuss the case 1, we will demonstrate that this case is impossible. For any $t \in [0, T^*)$, we can fix the values of $w$ for all edges except the one currently under consideration. At this point, $\kappa(t)$ can be expressed as a function of $w(t)$, i.e., $\kappa(t) = \kappa(w(t))$. According to $\kappa(t) > \kappa(0)$ and Lemma B.1, we know that $\kappa(w(t))$ increases monotonically as $w(t)$ decreases, i.e., $\frac{\partial \kappa(w(t))}{\partial w(t)} \leq -L < 0$. Please note that this holds for all $t \in [0, +\infty)$, not just within $[0, T^*)$. So for any $t \in [0, +\infty)$, we have:

$$\frac{\partial \kappa(t)}{\partial t} = \frac{\partial \kappa(w(t))}{\partial w(t)} \frac{\partial w(t)}{\partial t} = -\kappa(w(t))w(t)\frac{\partial \kappa(w(t))}{\partial w(t)} > 0. \tag{63}$$

Therefore:

$$\frac{\partial w(t)}{\partial t} = -\kappa(t)w(t) \leq -\kappa(0)\epsilon < 0, \quad \forall t \in [0, +\infty). \tag{64}$$

However, this is impossible because it would lead to $w(t)$ decreasing without bound, while $w(t) \geq \epsilon$. This results in a contradiction, so this situation cannot occur.

Next, we consider Case 2. Using a similar proof method as in Case 1, we can show that for $\kappa(t) > 0$, $\frac{\partial \kappa(w(t))}{\partial w(t)} \geq L > 0$ holds. Reviewing $w(t)$ over $[0, T^*)$, it is monotonically decreasing. To maximize $T^*$, we need to: 1) maximize $w(0) - w(T^*)$, where lead to $w(0) = 2 + \epsilon$ and $w(T^*) = \epsilon$. 2) minimize $|\frac{\partial w}{\partial t}|$. Since $\kappa(T^*) = \delta$, for any $t \in [0, T^*]$, the minimum value of $\kappa(t)$ satisfies $\kappa(t) - \kappa(T^*) = L(w(t) - w(T^*))$, so: $\kappa(t) = L(w(t) - \epsilon) + \delta$. Therefore we have:

$$\frac{\partial w(t)}{\partial t} = -[L(w(t) - \epsilon) + \delta]w(t) = -Lw(t)^2 + (L\epsilon - \delta)w(t). \tag{65}$$

This is a first-order nonlinear differential equation of the form $y' = -ay^2 + by$, whose general solution is $y = \frac{b}{Ce^{bx} - a} + \frac{b}{a}$, where $a = L$ and $b = L\epsilon - \delta$. By manipulating the general solution, we obtain:

$$Ce^{bt} = \frac{aw(t)}{aw(t) - b}. \tag{66}$$

When $t = T^*$, $w(t) = \epsilon$, and when $t = 0$, $w(t) = 2 + \epsilon$. Substituting these values, $T^*$ can be solved as follows:

$$\begin{aligned} T^* &= \frac{1}{b} \ln\left(\frac{aw(T)}{aw(T) - b} \cdot \frac{aw(0) - b}{aw(0)}\right) \\ &= \frac{1}{L\epsilon - \delta} \ln\left(\frac{2L + \delta}{\delta(2 + \epsilon)}\right). \end{aligned} \tag{67}$$

Treating $L$ and $\epsilon$ as non-zero constants, we have:

$$T^* = \mathcal{O}\left(\ln\left(\delta^{-1}\right)\right). \tag{68}$$

### B.4 DERIVATION OF PROPOSITION 4

From the condition $|\boldsymbol{h}_i(t)| \equiv 1$, we know that $\partial_t \|\boldsymbol{h}_i(t)\|^2 \equiv 0$, which expands using the chain rule to:

$$2 \left\langle \boldsymbol{h}_i(t), \frac{\partial \boldsymbol{h}_i(t)}{\partial t} \right\rangle = 0. \tag{69}$$

Moreover, due to $w_{ij}(t) = \boldsymbol{h}_i(t)^T \boldsymbol{h}_j(t) + 1 + \epsilon$, we know that $\frac{\partial w(\boldsymbol{h}_i, \boldsymbol{h}_j)}{\partial \boldsymbol{h}_i} = \boldsymbol{h}_j(t)$. To simplify notation, we omit the independent variable $t$ and denote $\frac{\partial \boldsymbol{h}_i(t)}{\partial t}$ as $\boldsymbol{y}$. Let $\mu = -\frac{\kappa_{ij}(t) w(\boldsymbol{h}_i, \boldsymbol{h}_j)}{1 + \lambda(\boldsymbol{h}_i(t), \boldsymbol{h}_j(t))}$. At this point, the original problem takes the following quadratic programming form:

$$\begin{aligned} \min \quad & \boldsymbol{y}^T \boldsymbol{y} \\ \text{s.t.} \quad & \boldsymbol{h}_j^T \boldsymbol{y} = \mu, \\ & \boldsymbol{h}_i^T \boldsymbol{y} = 0. \end{aligned} \tag{70} \tag{71}$$

Consider the method of Lagrange multipliers:

$$\begin{aligned} & \nabla_{\boldsymbol{y}} L(\boldsymbol{y}) = 0 \\ \longrightarrow \quad & \nabla_{\boldsymbol{y}} \left( \boldsymbol{y}^T \boldsymbol{y} + \lambda(\boldsymbol{h}_j^T \boldsymbol{y} - \mu) + \tau \boldsymbol{h}_i^T \boldsymbol{y} \right) = 0 \\ \longrightarrow \quad & 2\boldsymbol{y} + \lambda \boldsymbol{h}_j + \tau \boldsymbol{h}_i = 0. \end{aligned} \tag{72}$$

Left-multiplying Eq. (72) by $\boldsymbol{h}_i^T$, and combining it with Eq. (71) and $\boldsymbol{h}_i^T \boldsymbol{h}_i = 1$, we get:

$$\tau = -\lambda \boldsymbol{h}_i^T \boldsymbol{h}_j. \tag{73}$$

Similarly, left-multiplying Eq. (72) by $\boldsymbol{h}_j^T$:

$$\begin{aligned} & 2\boldsymbol{h}_j^T \boldsymbol{y} + \lambda + \tau \boldsymbol{h}_j^T \boldsymbol{h}_i = 0 \\ \longrightarrow \quad & 2\mu + \lambda - \lambda(\boldsymbol{h}_j^T \boldsymbol{h}_i)^2 = 0 \\ \longrightarrow \quad & \lambda = -2\mu \left( 1 - (\boldsymbol{h}_j^T \boldsymbol{h}_i)^2 \right)^{-1} \end{aligned} \tag{74}$$

Thus we can obtain the solution to this optimization problem:

$$\begin{aligned} \boldsymbol{y}^* &= -\frac{\lambda \boldsymbol{h}_j + \tau \boldsymbol{h}_i}{2} = -\frac{\lambda \boldsymbol{h}_j - \lambda \boldsymbol{h}_i^T \boldsymbol{h}_j \boldsymbol{h}_i}{2} = -\frac{\lambda \left( \boldsymbol{I} - \boldsymbol{h}_i \boldsymbol{h}_i^T \right) \boldsymbol{h}_j}{2} \\ &= \frac{\mu \left( \boldsymbol{I} - \boldsymbol{h}_i \boldsymbol{h}_i^T \right) \boldsymbol{h}_j}{1 - (\boldsymbol{h}_j^T \boldsymbol{h}_i)^2} = \frac{\mu}{1 - (\boldsymbol{h}_i^T \boldsymbol{h}_j)^2} \left[ \boldsymbol{h}_j - \cos(\boldsymbol{h}_i, \boldsymbol{h}_j) \boldsymbol{h}_i \right] = -\kappa_{ij}'(t) \left[ \boldsymbol{h}_j - \cos(\boldsymbol{h}_i, \boldsymbol{h}_j) \boldsymbol{h}_i \right]. \end{aligned} \tag{75}$$

Note that the constraint condition does not provide an upper bound for $\|\boldsymbol{y}\|$, but $\|\boldsymbol{y}\| \geq 0$ always holds. Therefore, $\boldsymbol{y}^*$ corresponds to the point of minimum $\boldsymbol{y}$.

### B.5 PROOF OF THEOREM 5

We first define an EdgeNet layer as follow:

$$e_{ij}^{(k+1)} = \text{MLP}_{\theta_2}^{(k)} \left( \sum_{u \sim i} \text{MLP}_{\theta_1}^{(k)} \left( e_{ui}^{(k)} \right) \Big\| e_{ij}^{(k)} \Big\| \sum_{v \sim j} \text{MLP}_{\theta_1}^{(k)} \left( e_{vj}^{(k)} \right) \right), \tag{76}$$

where, $e_{ij}$ represents the attributes on edges $i \sim j$, which can be a scalar or vector. $\|$ indicates concat operation. EdgeNet can be viewed as a natural generalization of DeepSet (Zaheer et al., 2017) on graphs, which performs a permutation-invariant mapping of the neighborhood of an edge.

> **Theorem B.9** (Formal version of Theorem 5). *Let $\lambda \equiv 1$. When Forman-Ricci curvature $\kappa^{\mathrm{FR}}$, Ollivier-Ricci curvature $\kappa^{\mathrm{OR}}$ or approximate resistance curvature $\widetilde{\kappa}^{\mathrm{RC}}$ are used as the definition of edge curvature in GNRF, there exists an EdgeNet that can approximate the aggregation weight of GNRF $\kappa'$ with arbitrarily high precision. Respectively, we have:*
>
> - *If $\kappa \equiv \kappa^{\mathrm{FR}}$, then a 1-layer EdgeNet with inputs $e_{ij}^{(0)} \equiv \boldsymbol{h}_i(t)\|\boldsymbol{h}_j(t)$ approximate $\kappa'$, i.e., $\kappa'_{ij}(t) = e_{ij}^{(1)}$.*
> - *If $\kappa \equiv \widetilde{\kappa}^{\mathrm{RC}}$, then a 2-layer EdgeNet with inputs $e_{ij}^{(0)} \equiv \boldsymbol{h}_i(t)\|\boldsymbol{h}_j(t)$ approximate $\kappa'$, i.e., $\kappa'_{ij}(t) = e_{ij}^{(2)}$.*
> - *If $\kappa \equiv \kappa^{\mathrm{OR}}$, then a 1-layer EdgeNet with inputs $e_{ij}^{(0)} \equiv \boldsymbol{h}_i(t)\|\boldsymbol{h}_j(t)\|\boldsymbol{v}_i$ approximate $\kappa'$, i.e., $\kappa'_{ij}(t) = e_{ij}^{(1)}$.*
>
> *Here, $\boldsymbol{v}_i$ is the $i$-th eigen-vector of the adjacent matrix $\boldsymbol{A}$.*

According to the standard Universal Approximation Theorem (Hornik et al., 1989), an MLP can approximate a continuous function to any desired accuracy. Therefore, to prove Theorem 5, we only need to assign each learnable MLP in the EdgeNet a continuous function. By approximating eachMLP to its corresponding continuous function, EdgeNet can ultimately approximate any defined curvature.

If $\lambda \equiv 1$, then there exist a continuous function $f$ such that:

$$\kappa'_{ij} = \frac{\kappa_{ij} w_{ij}}{2 - 2(w_{ij} - 1 - \epsilon)} \equiv f(\kappa_{ij}, w_{ij}). \tag{77}$$

For simplicity, we also denote $\mathrm{MLP}_{\theta_n}^{(k)}$ as $\varphi_n^{(k)}$. let $\varphi_1^{(0)}\left(\boldsymbol{h}_i(t)\|\boldsymbol{h}_j(t)\right) = w_{ij}(t)^{-0.5}$ and $\varphi_2^{(0)}(x\|y\|z\|r) = f(2 - (y^T z + 1 + \epsilon)(x + r), y^T z + 1 + \epsilon)$ then we approximate Forman-Ricci curvature:

$$
\begin{aligned}
e_{ij}^{(1)} &= \varphi_2^{(0)}\left(\sum_{u \sim i} \varphi_1^{(0)}\left(\boldsymbol{h}_u(t)\|\boldsymbol{h}_i(t)\right) \left\|\boldsymbol{h}_i(t)\|\boldsymbol{h}_j(t)\right\| \sum_{v \sim j} \varphi_1^{(0)}\left(\boldsymbol{h}_v(t)\|\boldsymbol{h}_j(t)\right)\right) \\
&= \varphi_2^{(0)}\left(\sum_{u \sim i} w_{ui}^{-\frac{1}{2}} \left\|\boldsymbol{h}_i(t)\|\boldsymbol{h}_j(t)\right\| \sum_{v \sim j} w_{vj}^{-\frac{1}{2}}\right) \\
&= \kappa_{ij}^{\mathrm{FR}}
\end{aligned}
$$

let $\varphi_1^{(0)}\left(\boldsymbol{h}_i(t)\|\boldsymbol{h}_j(t)\right) = w_{ij}(t)$, $\varphi_2^{(0)}(x\|y\|z\|r) = \left(\frac{1}{Nx} + \frac{1}{Nr}\right)\|y^T z + 1 + \epsilon$, $\varphi_1^{(1)}(x\|y) = xy$ and $\varphi_2^{(1)}(x\|y\|z\|r) = f(\frac{2 - (x+r)}{y}, z)$ then we approximate $\widetilde{\kappa}_{ij}^{\mathrm{RC}}$:

$$
\begin{aligned}
e_{ij}^{(1)} &= \varphi_2^{(0)}\left(\sum_{u \sim i} \varphi_1^{(0)}\left(\boldsymbol{h}_u(t)\|\boldsymbol{h}_i(t)\right) \left\|\boldsymbol{h}_i(t)\|\boldsymbol{h}_j(t)\right\| \sum_{v \sim j} \varphi_1^{(0)}\left(\boldsymbol{h}_v(t)\|\boldsymbol{h}_j(t)\right)\right) \\
&= \varphi_2^{(0)}\left(\sum_{u \sim i} w_{ui} \left\|\boldsymbol{h}_i(t)\|\boldsymbol{h}_j(t)\right\| \sum_{v \sim j} w_{vj}\right) \\
&= \widetilde{r}_{ij}\|w_{ij}
\end{aligned}
$$

$$e_{ij}^{(2)} = \varphi_2^{(1)} \left( \sum_{u \sim i} \varphi_1^{(1)} \left( \widetilde{r}_{ui} \| w_{ui} \right) \left\| \widetilde{r}_{ij} \right\| w_{ij} \left\| \sum_{v \sim j} \varphi_1^{(1)} \left( \widetilde{r}_{vj} \| w_{vj} \right) \right) \right.$$

$$= \varphi_2^{(1)} \left( \sum_{u \sim i} \widetilde{r}_{ui} w_{ui} \left\| \widetilde{r}_{ij} \right\| w_{ij} \left\| \sum_{v \sim j} \widetilde{r}_{vj} w_{vj} \right) \right.$$

$$= \widetilde{\kappa}_{ij}^{RC}$$

For $\kappa_{ij}^{\mathrm{OR}}$, the situation becomes a bit more complex. Noting $\kappa_{ij}^{\mathrm{OR}} = 1 - W_1(\mu_i, \mu_j)$, our objective transforms into approximating the 1-Wasserstein distance on the graph using neural networks. Due to the result of Lamma B.3, we need to find a representation for each node such that there exists a distance function on this representation to form a compact metric space. For Ollivier-Ricci Curvature, this distance must be the shortest path distance. To achieve this, we can to perform spectral decomposition on the unweighted adjacency matrix of graph $\mathcal{G}$ to obtain a set of eigenvalues and eigenvectors:

$$\{\boldsymbol{v}_i\}, \boldsymbol{\lambda} = \mathrm{SVD}(\boldsymbol{A}). \tag{78}$$

Then, according to Lamma B.2, there exists a function $f$ such that $f(\boldsymbol{v}_i, \boldsymbol{v}_j) = d(i, j)$, where $d(i, j)$ is the shortest path distance between nodes $i$ and $j$. Therefore, $\{\boldsymbol{v}_i\}$ forms a reasonable set of points in the graph shortest distance space. In this time, there exist trainable functions $\phi_1$, $\phi_2$, and $\phi_3$ such that:

$$W_1(\mu_i, \mu_j) = \phi_3 \left[ \phi_2 \left( \sum_{u \sim i} w_{ui} \phi_1(\boldsymbol{v}_u) \right) + \phi_2 \left( \sum_{v \sim j} w_{vj} \phi_1(\boldsymbol{v}_v) \right) \right]. \tag{79}$$

Let $\varphi_1^{(0)}(\boldsymbol{h}_i(t) \| \boldsymbol{h}_j(t) \| \boldsymbol{v}_i) = w_{ij}(t) \phi_1(\boldsymbol{v}_i)$ and $\varphi_2^{(0)}(x \| y \| z \| r \| s) = f(1 - \phi_3(\phi_2(x) + \phi_2(s)), y^T z + 1 + \epsilon)$ then we have:

$$e_{ij}^{(1)} = \varphi_2^{(0)} \left( \sum_{u \sim i} \varphi_1^{(0)} \left( \boldsymbol{h}_u(t) \| \boldsymbol{h}_i(t) \| \boldsymbol{v}_u \right) \left\| \left( \boldsymbol{h}_i(t) \| \boldsymbol{h}_j(t) \| \boldsymbol{v}_i \right) \right\| \sum_{v \sim j} \varphi_1^{(0)} \left( \boldsymbol{h}_v(t) \| \boldsymbol{h}_j(t) \| \boldsymbol{v}_v \right) \right)$$

$$= \varphi_2^{(0)} \left( \sum_{u \sim i} w_{ui} \phi_1(\boldsymbol{v}_u) \left\| \left( \boldsymbol{h}_i(t) \| \boldsymbol{h}_j(t) \| \boldsymbol{v}_i \right) \right\| \sum_{v \sim j} w_{vj} \phi_1(\boldsymbol{v}_v) \right)$$

$$= \kappa_{ij}^{\mathrm{OR}}$$

# C  EXPERIMENTS

## C.1  IMPLEMENT DETAILS

**Code.** An implementation is available at:

https://github.com/Loong-Chan/GNRF_new

**Models.** In the experiments, we set EdgeNet to be single-layer because we found that this performed well enough. Specifically, we perform experiments using the following formula:

$$\frac{\partial \boldsymbol{h}_i(t)}{\partial t} = \sum_{j \sim i} -\mathrm{EdgeNet}_{ij}(t) \left[ \boldsymbol{h}_j(t) - \cos\left( \boldsymbol{h}_j(t), \boldsymbol{h}_i(t) \right) \boldsymbol{h}_i(t) \right], \tag{80}$$

where

$$\mathrm{EdgeNet}_{ij}(t) \equiv \mathrm{MLP}_{\theta_2} \left( \sum_{u \sim i} \mathrm{MLP}_{\theta_1}(e_{ui}) \left\| e_{ij} \right\| \sum_{v \sim j} \mathrm{MLP}_{\theta_1}(e_{vj}) \right), \tag{81}$$

and $e_{ij} \equiv \boldsymbol{h}_i(t) \| \boldsymbol{h}_j(t)$. Both $\mathrm{MLP}_{\theta_1}$ and $\mathrm{MLP}_{\theta_2}$ are 2 layers. Among them, the number of their hidden neurons and the number of output neurons of $\mathrm{MLP}_{\theta_1}$ are consistent with the dimension of

---

**Algorithm 1** Solve GNRF with Forward difference method (PyTorch_Geometric style)

---

1: **Input:** Features $\boldsymbol{X}$, labels $\boldsymbol{y}$, edge index $\mathcal{E}$
2: $\boldsymbol{H} = \mathsf{pre\_transform}(\boldsymbol{X})$;
3: $sour\_idx, dest\_idx = \mathcal{E}$;
4: **for** $t = 0, 1, \cdots, T-1$ **do**
5:    $\boldsymbol{H}_{sour}(t) = \boldsymbol{H}(t)[sour\_idx]$; $\boldsymbol{H}_{dest}(t) = \boldsymbol{H}(t)[dest\_idx]$;
6:    **if** $use\ EdgeNet$ **then**
7:        $\boldsymbol{E}^{(0)} = \boldsymbol{H}_{sour}(t) \| \boldsymbol{H}_{dest}(t)$;
8:        **for** $l = 0, 1, \cdots, L-1$ **do**
9:            $\widetilde{\boldsymbol{E}}_{sour}^{(l)} = \mathsf{scatter}(\mathrm{MLP}_{\theta_1}^{(l)}\left(\boldsymbol{E}^{(l)}\right), sour\_idx)$;
10:          $\widetilde{\boldsymbol{E}}_{dest}^{(l)} = \mathsf{scatter}(\mathrm{MLP}_{\theta_1}^{(l)}\left(\boldsymbol{E}^{(l)}\right), dest\_idx)$;
11:          $\boldsymbol{E}^{(l+1)} = \mathrm{MLP}_{\theta_2}^{(l)}\left(\widetilde{\boldsymbol{E}}_{sour}^{(l)} \middle\| \boldsymbol{E}^{(l)} \middle\| \widetilde{\boldsymbol{E}}_{dest}^{(l)}\right)$
12:        **end for**
13:        $\boldsymbol{K} = \boldsymbol{E}^{(L)}$;
14:    **else**
15:        Compute curvatures $\boldsymbol{\kappa}(t)$ by $\boldsymbol{H}(t)$ and $\mathcal{E}$;
16:        $\boldsymbol{K} = \frac{\boldsymbol{\kappa}(t) \circ (\cos(\boldsymbol{H}_{sour}(t), \boldsymbol{H}_{dest}(t)) + 1 + \epsilon)}{2 - 2\cos^2(\boldsymbol{H}_{sour}(t), \boldsymbol{H}_{dest}(t))}$;
17:    **end if**
18:    $\widetilde{\boldsymbol{H}}(t) = \boldsymbol{H}_{dest}(t) - \cos\left(\boldsymbol{H}_{sour}(t), \boldsymbol{H}_{dest}(t)\right)\boldsymbol{H}_{sour}(t)$;
19:    $\boldsymbol{H}(t+1) = \boldsymbol{H}(t) - \eta \cdot \mathsf{scatter}\left(-\boldsymbol{K} \circ \widetilde{\boldsymbol{H}}(t), sour\_idx\right)$;
20: **end for**
21: $\boldsymbol{Z} = \mathsf{post\_transform}(\boldsymbol{H}(T))$;
22: Compute loss and back propagation via $\boldsymbol{Z}$ and $\boldsymbol{y}$.

---

$\boldsymbol{h}$. The output dimension of $\mathrm{MLP}_{\theta_1}$ is 1. One can also extend the output dimension of $\mathrm{MLP}_{\theta_2}$ to $|\boldsymbol{h}|$, which we call channel-wise curvature.

An explicit scheme of GNRF can be given using forward time difference:

$$\boldsymbol{h}_i' = \boldsymbol{h}_i - \eta \sum_{j \sim i} -\mathrm{EdgeNet}_{ij} \cdot \left[\boldsymbol{h}_j - \cos\left(\boldsymbol{h}_j, \boldsymbol{h}_i\right)\boldsymbol{h}_i\right], \tag{82}$$

where $\eta$ is step size. When people set a termination time $T$, this update process will be executed multiple times within $[0, T]$, eventually producing output $\boldsymbol{h}(T)$. The division of time slices is automatically performed by the ODE solver and is highly related to the solution algorithm.

**Experimental Platform.** Our code is implemented in Python 3.11.5, with the primary libraries being PyTorch 2.1.1, PyTorch Geometric 2.4.0, and Torchdiffeq 0.2.4. All experiments are conducted on a single NVIDIA 4090 GPU with with 40GB of VRAM.

**Hyperparameters.** We fine-tune GNRF within the hyperparameter search space, performing up to 100 trials on each dataset. The hyperparameter search space is as follows:

| Hyperparameters | Search Space | Distribution | Remark |
|---|---|---|---|
| learning rate | $[10^{-5}, 10^{-2}]$ | log-uniform | N/A |
| weight decay | $[10^{-6}, 10^{-3}]$ | log-uniform | N/A |
| dropout | $[0.01, 0.99]$ | uniform | N/A |
| hidden dim | $\{64, 128, 256\}$ | categorical | For Ogbn-arxiv, it is fixed at 64. |
| time | $[0.1, 10]$ | log-uniform | N/A |

Table 4: Hyperparameter Search Space

**Compute curvature via EdgeNet.** In GNRF, we use the output of EdgeNet as the value of $\kappa_{ij}(t)$:

$$\text{EdgeNet}_{ij}(t) = \kappa'_{ij}(t) = \frac{\kappa_{ij}(t)w(\boldsymbol{h}_i, \boldsymbol{h}_j)}{(1 + \lambda(\boldsymbol{h}_i(t), \boldsymbol{h}_j(t)))(1 - (\boldsymbol{h}_i^T\boldsymbol{h}_j)^2)}. \tag{83}$$

Where $\lambda$ is a scaling factor used to adjust the influence ratio of Attir-DRF on $\boldsymbol{h}_i$ and $\boldsymbol{h}_j$ for edge $i \sim j$. Currently, we set $\lambda \equiv 1$ by default, and the calculation formula for the learnable time-varying edge curvature is as follows:

$$\kappa_{ij}(t) = \frac{2 - 2\boldsymbol{h}_i^T\boldsymbol{h}_j}{\boldsymbol{h}_i^T\boldsymbol{h}_j + 1}\text{EdgeNet}_{ij}(t). \tag{84}$$

This equation is applied in Section 5.1.

**Homophily.** To comprehensively evaluate the model's performance, we pay particular attention to the diversity of the datasets when making our selections. As a result, we chose three homophilic graphs and five heterophilic graphs. The homophilic graphs exhibit a higher level of homophily in comparison, with homophily defined as follows:

$$\mathcal{H} = \frac{|\{(u,v) : (u,v) \in \mathcal{E} \wedge y_u = y_v\}|}{|\mathcal{E}|}. \tag{85}$$

This is also referred to as edge homophily ratio in the literature.

### C.2 MORE EXPERIMENTS

**Graph rewiring (Table 5).** Graph rewiring typically has a complexity of $\mathcal{O}(|\mathcal{V}|^2)$ (FOSR (Karhadkar et al., 2022)) or $\mathcal{O}(|\mathcal{V}||\mathcal{E}|)$ (SDRF (Topping et al., 2021)), which restricts its application to smaller datasets. In contrast, GNRF operates with a $\mathcal{O}(|\mathcal{V}| + |\mathcal{E}|)$ complexity while still offering similar curvature adjustments as graph rewiring. Moreover, graph rewiring overlooks label information, potentially losing valuable structural priors that are useful for downstream tasks, which hinder consistent improvements. The end-to-end GNRF effectively addresses this issue.

| Backbone | Rewiring | Cornell | Wisconsin | Texas |
|----------|----------|---------|-----------|-------|
| GCN | FOSR | 57.75($\uparrow$ 2.61) | 62.50($\uparrow$ 0.90) | 57.33($\downarrow$ 2.67) |
| | SDRF | 56.63($\uparrow$ 1.49) | 61.60($\uparrow$ 0.00) | 55.11($\downarrow$ 4.89) |
| GNRF | FOSR | 86.49($\downarrow$ 0.79) | 88.67($\uparrow$ 0.67) | 83.38($\downarrow$ 4.01) |
| | SDRF | 87.39($\uparrow$ 0.09) | 87.33($\downarrow$ 0.67) | 84.98($\downarrow$ 2.41) |

Table 5: GNRF as the backbone model for graph rewiring

**Graph classification (Table 6).** We report graph classification results on three commonly used molecular graph or protein graph datasets (NCI1, DD, PROTEINS). We use the widely adopted 80%/10%10% train/validation/test ratio for random splitting (Ma et al., 2019; Zhang et al., 2019; Ying et al., 2018). And report the mean and variance over 10 different divisions. We perform both Sum and Mean pooling for all methods. The parameter search range remains consistent with the main experiment (Table 1). We found that continuous-depth GNNs generally perform better. We speculate that this may be because the graph-level task requires fusing information from all node information in the entire graph, which is a challenge for discrete GNNs, but is easier for continuous GNNs. This is because in order to achieve sufficiently high accuracy, the ODE solver often needs to perform many time step within $[0, T]$, and it is usually much more than the common layer setting of discrete GNN (for example, within 5). GNRF performs better than the current advanced continuous depth GNN, namely ACMP.

**Long range graph learning (Table 7).** We verify the ability of GNRF to combat the over-squashing problem on two graph classification datasets with more than 1 million node reviews - Peptides-func and Peptides-struct. We use the same partitioning and verification methods as in Dwivedi et al. (2022). We use two commonly used position/structure encodings: LapPE and RWSE to enhance model performance Rampášek et al. (2022). GatedGCN Li et al. (2015) and SAN Kreuzer et al. (2021) were added for comparison. Based on our results, we find that GNRF shows significant

| Pooling | NCI1 | | DD | | PROTEINS | |
|---|---|---|---|---|---|---|
| | Sum | Mean | Sum | Mean | Sum | Mean |
| GCN+res | $75.28 \pm 1.33$ | $76.26 \pm 1.05$ | $74.81 \pm 0.96$ | $76.12 \pm 0.57$ | $75.42 \pm 1.30$ | $75.82 \pm 0.35$ |
| GAT+res | $73.25 \pm 2.11$ | $73.65 \pm 1.35$ | $76.68 \pm 0.88$ | $77.26 \pm 2.01$ | $74.44 \pm 1.35$ | $74.51 \pm 0.96$ |
| GRAND | $76.54 \pm 1.51$ | $77.82 \pm 0.68$ | $76.56 \pm 0.55$ | $78.51 \pm 0.87$ | $77.12 \pm 0.53$ | $78.25 \pm 1.14$ |
| ACMP | $77.42 \pm 0.60$ | $79.09 \pm 0.77$ | $75.82 \pm 1.83$ | $78.44 \pm 0.53$ | $78.88 \pm 0.33$ | $78.34 \pm 0.66$ |
| GNRF | $79.59 \pm 0.69$ | $81.67 \pm 0.51$ | $78.52 \pm 0.64$ | $79.08 \pm 0.88$ | $78.59 \pm 2.12$ | $80.12 \pm 0.54$ |

Table 6: We compare GNRF with other models on 3 graph classification datasets.

improvements over classic message-passing-based GNNs. Without additional encoding, GNRF improves performance by at least 3% over GCN on both Peptides-func and Peptides-struct. When additional encodings are used, GNRF's performance can rival that of SAN (a Transformer-based architecture). However, we acknowledge that GNRF still struggles to match the state-of-the-art Graph Transformer methods on the LRGB dataset. But it is forgivable because GNRF remains a fully message-passing architecture, where first-order neighbors are the only direct source of information for feature updates. Compared to Graph Transformer methods, GNRF has much lower computational complexity and is more suitable for large-scale single-graph scenarios.

| | GCN | GatedGCN+RWSE | SAN+LapPE | SAN+RWSE | GNRF | GNRF+LapPE | GNRF+RWSE |
|---|---|---|---|---|---|---|---|
| Peptides-func AP($\uparrow$) | 0.5930±0.0023 | 0.6069±0.0035 | 0.6384±0.0121 | 0.6439±0.0075 | 0.6233±0.0080 | 0.6455±0.0062 | 0.6480±0.0056 |
| Peptides-struct MAE($\downarrow$) | 0.3496±0.0013 | 0.3357±0.0006 | 0.2683±0.0043 | 0.2545±0.0012 | 0.3166±0.0053 | 0.2675±0.0044 | 0.2811±0.0031 |

Table 7: We verify GNRF on Long Range Graph Benchmark.

**Model depth (Table 8).** We measure the resource consumption and performance of the models during training on the OGBN-Arxiv and OGBN-Year datasets. Since these two datasets share the same graph structure and attributes (differing only in labels), we only present the resource consumption results for OGBN-Arxiv. The length of the hidden representation for all models was fixed at 64. Two classic deep GNNs: APPNP (Gasteiger et al., 2018) and GCNII (Chen et al., 2020b), are used for comparison. In GCNII, each layer of discrete-depth GNNs uses different learnable parameters, resulting in parameter count, memory, and time consumption increasing linearly with the number of layers. In contrast, continuous-depth GNNs have parameter count and memory usage independent of depth, giving them a significant advantage at greater depths. Compared with ACMP, the training time of GNRF does not increase as the depth increases. This is an advantage brought by the fixed-step ODE solver. In terms of performance, whether it is the homophilious graph (Arxiv) or the hetrophilious graph (Year), coutinious-depth GNNs (In particular, GNRF) shows significant advantages when the depth is 4 or 16. When the depth becomes deeper, the performance of both ACMP and GNRF decreases due to the accumulation of errors in the ODE solver.

| | | GCNII | | | | APPNP | | | | ACMP | | | | GNRF | | |
|---|---|---|---|---|---|---|---|---|---|---|---|---|---|---|---|---|
| | | #Param | Mem. | Time | Acc. | #Param | Mem. | Time | Acc. | #Param | Mem. | Time | Acc. | #Param | Mem. | Time | Acc. |
| Arxiv & Year | 4 | 27.2k | 2.24G | 0.14s | 63.55 / 33.94 | 15.0k | 1.64G | 0.16s | 64.52 / 39.38 | 19.5k | 7.15G | 8.43s | 67.16 / 47.55 | 35.9k | 11.5G | 0.79s | **69.25** / **48.55** |
| | 16 | 76.0k | 4.06G | 0.24s | 64.37 / 35.01 | 15.0k | 1.64G | 0.22s | 64.51 / 39.03 | 19.5k | 7.15G | 13.7s | **65.72** / 43.53 | 35.9k | 11.5G | 0.79s | 65.14 / **44.13** |
| | 64 | 273k | 10.7G | 0.66s | **67.55** / 35.44 | 15.0k | 1.64G | 0.55s | 64.66 / 39.08 | 19.5k | 7.15G | 17.6s | 51.72 / **42.31** | 35.9k | 11.5G | 0.79s | 55.23 / 40.15 |

Table 8: We verify the performance of GNRF at different depths and report the model overhead.

## C.3 FUTURE DIRECTION

In this paper, we focus on the application of Attribute Discrete Ricci flow in message passing-based discrete/continuous-depth GNNs. However, in view of the excellent performance of Graph Transformers (GTs), especially on graph-level tasks, we believe that it is also meaningful to consider the application of Attri-DRF in this type of method. We believe that this generalization may be feasible, based on two observations:

(1) In theory, there are certain curvatures that can be defined on any node pair $(i, j)$ without requiring $i$ and $j$ to be adjacent (For example, Ollivier Ricci curvature). This is in GTs is very useful because GTs directly aggregates information from the entire graph.

(2) In practice, a significant difference between our model GNRF and GARND is that the aggregation weight replaces the attention coefficient with a curvature-aware coefficient. Given the widespread reference of attention coefficients in GTs, this replacement is likely natural.

We also provide a possible promotion here. Let $\mathsf{PE}(\cdot)$ be some position encoding function and $\mathsf{sim}(\cdot, \cdot)$ be some similarity function. We can let $w_{ij}(t) \equiv \mathsf{sim}(\mathsf{PE}(i, t), \mathsf{PE}(j, t))$ to get the generalization of Attri-DRF:

$$\frac{\partial \mathsf{sim}(\mathsf{PE}(i, t), \mathsf{PE}(j, t))}{\partial t} = -\kappa_{ij}(t)\mathsf{sim}(\mathsf{PE}(i, t), \mathsf{PE}(j, t)). \tag{86}$$

We leave application on GTs of this definition to future work.

