# OpenReview forum: "Graph Neural Ricci Flow: Evolving Feature from a Curvature Perspective"
_ICLR.cc/2025/Conference — ICLR 2025 Poster_

### Official Review · Reviewer_1v62 · 2024-10-27

**Soundness:** 3
**Presentation:** 2
**Contribution:** 3
**Rating:** 6
**Confidence:** 4

**Summary:**

The paper introduces Graph Neural Ricci Flow (GNRF), a novel method for evolving node features in Graph Neural Networks (GNNs) using a differential equation-inspired approach based on the Ricci flow. The authors generalize the Discrete Ricci Flow to attributed graphs, where each edge's curvature converges toward zero over time. This has two key consequences: it bounds the graph's Dirichlet energy and provides a data-independent curvature decay rate. GNRF is unique because it computes time-varying curvature efficiently in linear time, unlike traditional curvature-based methods, which are typically precomputed and limited to specific curvature definitions.

The motivation behind the GNRF stems from the limitations of existing GNNs that rely on heat diffusion equations, which often lead to over-smoothing. Instead, the paper explores an alternative differential equation—Ricci flow—to mitigate over-smoothing and create more stable, non-smooth node representations. This innovative approach contrasts with traditional methods that view curvature as a static, precomputed property tied only to graph topology. By allowing curvature to evolve with node features, GNRF enables more dynamic and flexible graph learning.

**Strengths:**

- Novelty: The paper introduces an interesting and innovative approach with GNRF, applying Discrete Ricci Flow to attributed graphs in a novel way. By allowing edge curvature to evolve dynamically with node attributes, GNRF moves beyond the limitations of traditional methods that rely on static, precomputed curvature. Its ability to work with any curvature definition and to compute curvature in linear time addresses key concerns around scalability and efficiency. This flexibility offers a practical and effective way to handle common challenges such as over-smoothing and over-squashing in graph neural networks. Overall, GNRF presents a meaningful advancement in curvature-aware graph learning.
- Theoretical results: The paper offers solid theoretical contributions that help establish the soundness of the Attribute Discrete Ricci Flow framework. One key result is the demonstration that edge curvature naturally converges toward zero, ensuring a stable evolution of node features and addressing potential issues like over-smoothing and over-squashing. The bounding of the Dirichlet energy provides additional assurance that node representations maintain a balance between being too homogeneous or too distinct.

**Weaknesses:**

- Limited experimental results: The paper's experimental evaluation has some limitations. The focus is exclusively on node classification tasks, raising the question of why the method wasn't tested on other common tasks like graph classification or regression, which would provide a broader view of its applicability. Additionally, of the seven node classification datasets used, three (Cornell, Wisconsin, and Texas) are notably small, making it difficult to draw definitive conclusions about the method’s performance on more challenging or larger-scale data. Furthermore, on two of the larger datasets (Cora_Full and PubMed), the proposed method performs only within the statistical margin of error compared to the baselines, which limits its ability to demonstrate a clear and significant improvement over existing methods. Overall, I believe the paper would benefit significantly if the authors added some experimental results on graph classification/ regressions tasks, for example the LRGB datasets [1].
- Baseline comparisons: The paper’s comparison with baseline methods raises some concerns regarding its evaluation methodology. Specifically, on the Tolokers and Roman Empire datasets, the authors use a 60/20/20 train/validation/test split, but the results reported for baseline models like GCN are significantly lower than what is found in the original work, which used a 50/25/25 split. When compared with the results in “A Critical Look at the Evaluation of GNNs under Heterophily” (2023), the proposed method (82.55 on Tolokers) does not seem to outperform a simple baseline like GCN on Tolokers (83.64), raising questions about whether the method truly offers improvements in these settings.

Overall, I would be happy to increase my score to a 5 or 6 if the authors can convincingly address the above two points and show the practical usefulness of their method.

[1] Dwivedi, Vijay Prakash, et al. "Long range graph benchmark." Advances in Neural Information Processing Systems 35 (2022): 22326-22340.

**Questions:**

Could the authors explain why they are only using the Tolokers and Roman Empire datasets from “A Critical Look at the Evaluation of GNNs under Heterophily” and not the three other node classification datasets?

---

> ### Author Response · Authors · 2024-11-14
>
> Dear Reviewer,
>
> Thank you for your detailed review and for recognizing the innovativeness of our approach. We also acknowledge your concerns regarding the limitations of our experimental results, and we would like to address them as follows:
>
> ## Regarding Limited Experimental Results
> In the original manuscript, we aimed to provide a more diverse set of experiments to offer a comprehensive understanding of our method. However, we acknowledge that the experimental results on the main task (node classification) were somewhat limited. To address this, we have added the remaining three datasets from HeterophilousGraphs: **Minesweeper**, **Questions**, **Amazon-ratings**, and two datasets from CitationFull: **DBLP** and **Cora_ML** as supplementary results, which can be found below. Additionally, in response to other reviewers' requests, we will report results on the **OGBN-Arxiv** and **OGBN-Year** datasets (both with over 100k nodes) in the coming days. Lastly, regarding the LRGB dataset, experiments are ongoing, and we will report results on **Peptides-func** and **Peptides-struct** shortly. We appreciate your patience.
>
> ## Regarding Baseline Comparisons
> We revisited the paper and code from [1] and identified two key factors:
>
> 1. We reported results on Tolokers based on **accuracy**, whereas the original paper used **ROC-AUC**. Cross-metric comparisons are inappropriate, and we acknowledge that ROC-AUC is a better metric for binary classification. We will re-evaluate the results on Tolokers and update the paper accordingly. You will be notified once the updated results are available.
>
> 2. We noticed that in [1], they included **residual connections and an additional linear layer** for GCN (referred to as GCN+Res), while we reported results for the **vanilla GCN**. We found that residual connections had a significant effect on the HeterophilousGraphs benchmark but did not improve performance on DBLP and Cora_ML. Given that residual connections are designed for layered neural networks, we have not incorporated this module into our continuous deep neural network based on differential equations. Therefore, we feel that considering GCN+Res as a baseline for continuous deep GNNs may not be entirely fair. Nonetheless, our model (GNRF) still performs competitively.
>
> ## Regarding the Choice of Tolokers and Roman Empire
> This choice was random. We have now included the remaining datasets from our benchmark, so this should no longer be a concern.
>
> |               | Minesweeper         | Questions          | A.-ratings        | DBLP              | Cora_ML          |
> |---------------|----------------------|--------------------|-------------------|-------------------|------------------|
> | **GCN**       | 74.79 ± 1.78         | 50.21 ± 2.24      | 37.99 ± 0.61     | 83.93 ± 0.34     | 87.07 ± 1.21     |
> | **GCN+Res**   | 90.13 ± 0.70         | 75.45 ± 2.31      | 48.17 ± 0.55     | 82.64 ± 0.51     | 85.62 ± 0.72     |
> | **GRAND**     | 80.56 ± 3.12         | 54.90 ± 2.12      | 37.53 ± 0.36     | 84.60 ± 0.99     | 88.49 ± 0.81     |
> | **GNRF**      | 95.03 ± 0.20         | 73.86 ± 1.18      | 47.89 ± 1.08     | 85.73 ± 0.76     | 89.18 ± 0.19     |
>
> The above improvements will be updated in the paper soon. Once again, thank you for your professional review.
>
> [1] A Critical Look at the Evaluation of GNNs under Heterophily” (2023)

---

> ### Author Response · Authors · 2024-11-22
>
> Dear reviewers, all planned changes have now been included in our latest version of the paper. In particular, we conduct richer experiments to enable readers to more comprehensively evaluate the performance of GNRF. They can be found in Tables 1-3 in the main text of the paper and Tables 5-7 in Appendix C.2. At the same time, I also excerpt and record it for you:
>
> (Table 5) We first performed experiments on three commonly used graph classification data sets. Our experimental results were based on 80%/10%/10% division (after our research, this is a commonly used ratio), and reported 10 results. We found that the effect based on continuous depth GNN is generally better than the classic model. We speculate that this may be because the graph-level task requires fusing information from all node information in the entire graph, which is a challenge for discrete GNNs, but is easier for continuous GNNs. This is because in order to achieve sufficiently high accuracy, the ODE solver often needs to perform many time step within [0, T], and it is usually much more than the common layer setting of discrete GNN (for example, within 5). GNRF performs better than the current advanced continuous depth GNN, namely ACMP.
>
> | Pooling | NCI1         | NCI1           | DD     |DD         | PROTEINS           | PROTEINS     |
> |---------|--------------|--------------|--------------|--------------|--------------|-------------|
> |         | Sum          | Mean         | Sum          | Mean         |Sum          | Mean         |
> | GCN+res | 75.28 ± 1.33 | 76.26 ± 1.05 | 74.81 ± 0.96 | 76.12 ± 0.57 | 75.42 ± 1.30 | 75.82 ± 0.35 |
> | GAT+res | 73.25 ± 2.11 | 73.65 ± 1.35 | 76.68 ± 0.88 | 77.26 ± 2.01 | 74.44 ± 1.35 | 74.51 ± 0.96 |
> | GRAND   | 76.54 ± 1.51 | 77.82 ± 0.68 | 75.56 ± 0.55 | 78.51 ± 0.87 | 77.12 ± 0.53 | 78.25 ± 1.14 |
> | ACMP    | 74.42 ± 0.60 | 79.09 ± 0.77 | 75.82 ± 1.83 | 78.44 ± 0.53 | 78.88 ± 0.33 | 78.34 ± 0.66 |
> | GNRF    | 79.59 ± 0.69 | 81.67 ± 0.54 | 78.52 ± 0.64 | 79.08 ± 0.88 | 78.59 ± 2.12 | 80.12 ± 0.54 |
>
> (Table 6) According to your request, we have supplemented two datasets from the Long Range Graph Benchmark (LRGB) in Table 7. Our dataset partitioning and statistical methods are fully consistent with the official LRGB. Additionally, we directly cite data from the LRGB LeaderBoard for comparison to ensure the fairness of the results. Following the convention of LRGB, we also tested the gains provided by GNRF after using two common positional/structural encodings: LapPE and RWSE. Based on our results, we find that GNRF shows significant improvements over classic message-passing-based GNNs. Without additional encoding, GNRF improves performance by at least 3% over GCN on both Peptides-func and Peptides-struct. When additional encodings are used, GNRF's performance can rival that of SAN (a Transformer-based architecture). However, we acknowledge that GNRF still struggles to match the state-of-the-art Graph Transformer methods on the LRGB dataset. Nevertheless, we believe this is forgivable because GNRF remains a fully message-passing architecture, where first-order neighbors are the only direct source of information for feature updates. Compared to Graph Transformer methods, GNRF has much lower computational complexity and is more suitable for large-scale single-graph scenarios.
>
>
> |           | GCN                | GatedGCN+RWSE       | SAN+LapPE           | SAN+RWSE            | GNRF                | GNRF+LapPE          | GNRF+RWSE           |
> |-----------|--------------------|---------------------|---------------------|---------------------|---------------------|---------------------|---------------------|
> | Peptides-func AP(↑) | 0.5930±0.0023      | 0.6069±0.0035       | 0.6384±0.0121       | 0.6439±0.0075       | 0.6233±0.0080         | 0.6455±0.0062       | 0.6480±0.0056       |
> | Peptides-struct MAE(↓) | 0.3496±0.0013      | 0.3357±0.0006       | 0.2683±0.0043       | 0.2545±0.0012       | 0.3166±0.0053       | 0.2675±0.0044       | 0.2811±0.0031       |
>
> We hope that the additional experiments will address your concerns. We also look forward to your feedback to help us further improve the paper.

---

> > ### Comment · Reviewer_1v62 · 2024-11-22
> >
> > I would like to thank the authors for addressing my concerns and especially for providing a large number of additional experimental results. I find the results convincing and will therefore adjust my score accordingly.

---

> ### Author Response · Authors · 2024-11-22
>
> We are so grateful that the reviewer recognized our efforts! We will continue to improve our paper in the future!

---

### Official Review · Reviewer_QrP8 · 2024-10-30

**Soundness:** 3
**Presentation:** 3
**Contribution:** 3
**Rating:** 6
**Confidence:** 3

**Summary:**

The paper proposes a continuous GNN dynamics, namely GNRF by incorporating curvature based on the Ricci flow. In particular, by expressing edge weights as a function of node features, GNRF propagate features following a discrete (graph) Ricci flow. In order to avoid the costly computation of Ricci curvature on graphs, the paper proposes an auxiliary network for modeling curvature and learned end-to-end.  The paper provides several theoretical guarantees in terms of bounded Dirichlet energy and fast curvature decay. The experiments support the effectiveness of the method.

**Strengths:**

1. Compared to curvature-based graph rewiring, it is interesting and natural to incorporate Ricci curvature into the propagation of node features.

2. Theoretical developments are supportive of the claims.

**Weaknesses:**

1. It is unclear how EdgeNet approximates edge curvatures? In particular, given there are trainable parameters and in the experiments, EdgeNet is trained end-to-end with supervision only from the task, instead of actual curvature. How to ensure EdgeNet approximates the curvature in this case?

2. Theorem 5 is unclear. Does this mean there exists some network \phi_1, \phi_2 such that the network can approximate any curvature? Please give more explanations.

3. Even though the theory is well-developed, the main GNN algorithm in (14) seems to resemble GRAND, especially the EdgeNet seems to act like a re-weighting term as in graph attention. How does EdgeNet differ to the graph attention module? Can you add experiments to verify the difference?

**Questions:**

1. In Line 285, the paper claims the sign of EdgeNet_ij aligns with the sign of k_ij. I am not sure how this is achieved without the supervision from the actual curvature.

2. In Line 175 of Theorem 2, w_ij should be changed to k_ij?

3. In Section 5.1, the curvature seems to be computed from the EdgeNet? What about the actual curvature?

4. I am also curious whether there could be improvements when EdgeNet is replaced with actual curvature? This could be part of ablation study.

---

> ### Author Response · Authors · 2024-11-19
>
> Dear Reviewer,
>
> Thank you very much for your professional review comments. We have noticed that your main concerns focus on the model design, especially regarding EdgeNet. Below is our detailed response:
>
> ## Weakness 1
> Indeed, EdgeNet does not use a specific curvature definition. This is because, although there are multiple ways to define curvature, as far as we know, there is no theoretical guidance on how to choose the appropriate one in practice. Furthermore, based on experimental data from existing literature ([1], [2]) as well as additional experiments we conducted, we observed that the impact of using different curvature definitions on model performance is quite significant, yet no generalizable guidelines can be formed. Therefore, we believe that using an adaptive curvature definition may be a better choice. Additionally, as shown in Section 5.2 of the revised paper, even though EdgeNet does not rely on a specific curvature, it still exhibits behavior consistent with Ricci flow. This is because our theorem itself is independent of any particular curvature, providing a theoretical foundation for the introduction of EdgeNet.
>
> ## Weakness 2
> Your understanding is mostly correct. We implemented $\phi_1$ and $\phi_2$ as a two-layer MLP. In Theorem 5, we state that it is always possible to find suitable parameters for these MLPs such that the network output can approximate any curvature (i.e., the network structure is fixed). You can refer to Appendix B.5 in the revised paper for a rigorous proof of Theorem 5, as well as Appendix C.1 for the implementation details of EdgeNet.
>
> ## Weakness 3
> In the original version, we provided an intuitive and experimental discussion on the difference between GNRF and GRAND, and we have now further deepened this discussion. One major distinction lies in the sign of the aggregation weights. GRAND is derived from a heat diffusion model, resulting in all positive aggregation weights (i.e., the attention coefficients are always positive). In contrast, GNRF allows negative weights—specifically, when an edge has positive curvature, we use negative weights, and vice versa for negative curvature. This leads to fundamentally different behavior between GNRF and GRAND. While GRAND tends to smooth all node pairs, GNRF only smooths negative curvature node pairs while repelling positive curvature ones. In experiments, we demonstrated through ablation studies the effects when GNRF and GRAND differ only in aggregation weights. We found that GRAND performs poorly on heterophilious graphs (such as Tolokers and Roman-Empire), supporting our view on the importance of negative weights/attention coefficients.
>
> ## Question 1
> The original statement was indeed not precise, and we have corrected it. What we meant was that the aggregation weights of GNRF ($\kappa^\prime$) have the same sign as the curvature ($\kappa$), and EdgeNet’s role is to approximate $\kappa^\prime$. As mentioned in our response to Weakness 1, when using EdgeNet, the model actually utilizes a dataset-specific personalized curvature rather than a pre-defined curvature. The experiments in Section 5.2 of the revised paper confirm that this approach is feasible—using EdgeNet still adheres to the characteristics of Ricci flow, and our theoretical results also apply to EdgeNet.
>
> ## Question 2
> Yes, you are correct. This was indeed a typographical error, which we have fixed in the latest version of the paper. We have also updated the statements of all theorems with more detailed descriptions to ensure their rigor.
>
> ## Question 3
> Dear reviewer, please refer to our responses to Weakness 1 and Question 1.
>
> ## Question 4
> We have added this experiment in the latest version of the paper. Specifically, we replaced the curvature in GNRF with two real curvatures: Forman-Ricci Curvature and approximate resistance curvature, resulting in two new models: GNRF_FRC and GNRF_ARC. We validated these models on 12 datasets, and although the two variants have their own strengths and weaknesses, they generally perform worse than GNRF with EdgeNet, which further supports our belief that adaptive curvature is more advantageous.
>
> We hope that our response effectively addresses your concerns, and we are more than willing to provide further details on any other questions you may have and update the paper accordingly. Once again, thank you for your valuable feedback and suggestions!
>
>
> [1] Curvature filtrations for graph generative model evaluation
> [2] Curvature constrained mpnns: Improving message passing with local structural properties

---

> ### Author Response · Authors · 2024-11-22
>
> Dear Reviewer, we have now completed all the planned revisions. You can find details of these changes in the Official Comment and the latest version of the paper. We are eagerly awaiting your positive feedback.

---

> ### Author Response · Authors · 2024-11-23
> **Additional comments**
>
> We would like to provide a more detailed explanation regarding the three weaknesses you mentioned.
>
> ## Weakness 1 (On how GNRF approximates edge curvature)
> As explained in the previous comment, EdgeNet does not directly approximate any specific real-world definition of curvature during end-to-end training; rather, it acts as a dataset-adaptive curvature proxy. We support this approach both theoretically and experimentally. Theoretically, our results do not depend on any specific definition, and experimentally, we found that (1) using a specific curvature definition does not consistently perform well across all datasets (as shown in the figure below), and (2) even when using adaptive curvature, GNRF's performance aligns with that of Ricci flow (as shown in Sections 5.2 and 5.3 of the paper).
>
> | Dataset    | Corn. | Wisc. | Texas | R. Emp. | Tolo. | Mine. | Ques. | A.-rat. | C._Full | PubM. | DBLP | C._ML |
> |------------|-------|-------|-------|---------|-------|-------|-------|---------|---------|-------|------|-------|
> | GNRF       | 87.28 | 88.00 | 87.39 | 86.25   | 83.96 | 95.03 | 73.86 | 46.89   | 72.12   | 90.37 | 85.73| 89.18 |
> | GNRF_FRC   | 85.59 | 84.00 | 82.08 | 75.23   | 76.17 | 81.61 | 61.78 | 41.22   | 67.51   | 88.96 | 82.55| 87.29 |
> | GNRF_ARC   | 86.49 | 88.00 | 81.90 | 76.52   | 78.14 | 87.25 | 64.55 | 41.74   | 70.17   | 88.21 | 83.33| 89.43 |
>
> ## Weakness 2 (On the explanation of Theorem 5)
> We know that there are many definitions of curvature for edges in a graph. We found that there is a network architecture (EdgeNet) where, when a specific curvature definition (e.g., Forman-Ricci Curvature or others) is specified, we can always find appropriate parameters for this EdgeNet, such that it takes the neighborhood information of an edge as input and outputs the Forman-Ricci Curvature value. As shown in Appendix C.1, EdgeNet is actually composed of several MLPs, and its ability to approximate curvature comes from the universal approximation theorem of MLPs.
>
> ## Weakness 3 (On the difference between GNRF and GRAND)
> The neighbor aggregation weights in GRAND are actually attention coefficients, meaning they satisfy two constraints: normalization and non-negativity. However, for GNRF, the aggregation weights do not have these constraints; they can be negative, and negative weights yield significant benefits in heterophilious graphs. Another point is that attention coefficients come with an implicit bias: node pairs with similar features often receive higher weights. While this bias is often shown to be beneficial, we found that removing it can lead to unexpected results. As shown in Figure 5 of the main text, we observed that GNRF tends to reject pairs of nodes that are very similar, which in turn leads to smoother boundaries, exhibiting behavior that is quite different from that of GRAND. Finally, we present results from an ablation study. Here, \(d\) denotes the damping factor, and the difference between the models GRAND+d and GNRF lies only in the aggregation weight calculation. We observed that GNRF significantly outperforms GRAND+d, particularly on heterophilious graphs (Roman-Empire and Tolokers).
>
> | | Roman-Empire | Tolokers | Cora Full |
> |---|---|---|---|
> | GRAND | 60.12 | 79.01 | 67.66 |
> | GRAND+d | 58.57 | 78.78 | 67.31 |
> | GNRF | 86.26 (+26.14) | 83.96 (+4.95) | 72.12 (+4.46) |

---

> > ### Comment · Reviewer_QrP8 · 2024-11-24
> > **Thank you for the response**
> >
> > I thank the authors for providing the detailed responses. My main concerns are well addressed and thus I have increased the score accordingly.

---

> > > ### Author Response · Authors · 2024-11-24
> > >
> > > We are so grateful that the reviewer recognized our efforts! We will continue to improve our paper in the future!

---

### Official Review · Reviewer_tPnw · 2024-11-06

**Soundness:** 3
**Presentation:** 3
**Contribution:** 2
**Rating:** 6
**Confidence:** 4

**Summary:**

his paper introduces Graph Neural Ricci Flow (GNRF), designed to model a dynamic system called Attribute Discrete Ricci Flow (Attri-DRF) on graph.
Unlike traditional GNNs, which use multiple layers and pass outputs from one layer as inputs to the next, the model in this work employs only a single layer.  Instead, it iteratively updates node features and curvatures—treated similarly to edge weights—over discrete time steps according to the Attri-DRF ODE.

**Strengths:**

1. The model dynamically learns the curvature instead of relying on precomputed values, enabling it to adapt in response to both internal hidden features and the graph’s topology.
2. This approach is closely aligned with the heat flow equation.
3. As shown in Figure 3, the proposed framework achieves stable curvature over sufficient time steps. At the same time, the curvature concentrates around zero that can facilitate smoother information flow across the graph.

**Weaknesses:**

1. The network architecture used in this framework is relatively general. And it would be valuable to discuss the potential benefits of using more complex GNN architectures. Moreover, exploring the motivation behind the proposed framework with other related works [1, 2, 3] that focus on graph curvatures could provide additional insights.

2. The experiment primarily focuses on node classification tasks on small-scale graphs. To better validate the effectiveness of the proposed framework, applying it to larger graphs would be beneficial. For example, the ogbn-arxiv dataset could serve as a graph classification dataset with GNNs as baselines. Additionally, for non-homophilous graph datasets, larger datasets and relevant baselines are available in [4].

3. An efficiency study would be helpful. The computational cost of applying the ODE method should be explicitly discussed so readers can better understand its applicability. For instance, comparing parameters, training time, and GPU memory usage between this approach and other GNNs, such as GCN and GAT, would clarify its potential advantages and trade-offs.

[1] Curvdrop: A ricci curvature based approach to prevent graph neural networks from over-smoothing and over-squashing.
[2] Curvature Graph Neural Network.
[3] Hyperbolic variational graph neural network for modeling dynamic graphs
[4] Large Scale Learning on Non-Homophilous Graphs: New Benchmarks and Strong Simple Methods

**Questions:**

1. Could you share your thoughts on the relationship and differences between the curvature predicted in this architecture and edge attention mechanisms? And do you think it is possible to apply attention mechanism in the framework?

2. In EdgeNet, are the edge features from the previous time step used as input for the current time step? Equation 13 suggests that the previous edge features should be used, but the code appears to rely only on the previous node features without incorporating the prior edge features.

3. What is the formulation for updating node features at each subsequent time step? As a suggestion, including an illustrative figure or a pseudo-algorithm would help readers gain a clearer understanding of the overall framework.

4. While there is only one layer in the implementation, is it possible to apply a multiple layer GNN? And is it possible to connect layers with time steps in this case?

---

> ### Author Response · Authors · 2024-11-19
> **(1/N)**
>
> Dear Reviewer,
>
> Thank you for your professional comments and your recognition of our work. We have also noted your concerns, and below is our detailed response:
>
> ## Weakness 1
> We have comprehensively updated the paper to include more in-depth discussions. Specifically, we have added discussions on curvature-based edge sampling [1] and weighted aggregation [2] to provide a more holistic understanding of curvature graph learning. Additionally, other branches of Riemannian graph learning, such as hyperbolic graph learning [3], are discussed in the related work section of the appendix.  In addition, we have added a new appendix section C.3 to discuss the potential application of our proposed Attri-DRF in other GNN architectures, especially Graph Transformer. You can see our detailed response to this in reply (4).
>
> ## Weakness 2
> In the latest version, we have added 5 more datasets to the main experiments, bringing the total to 12, with the largest containing nearly 50,000 nodes. Additionally, for the resource overhead experiments, we have evaluated two larger datasets, OGBN-Arxiv and OGBN-Year [4], both of which have over 100,000 nodes. The results indicate that our method still shows stable improvements on these larger datasets.
>
> ## Weakness 3
> We have now supplemented the relevant experiments. In the resource overhead evaluation, we additionally report the number of trainable parameters, peak memory usage, and average training time per epoch. The results demonstrate that GNRF achieves a good balance across multiple metrics, particularly in deep model settings, where it incurs less overhead compared to traditional models like GCN. Furthermore, we have added a discussion on computational complexity in the main text. Based on widely recognized computation methods, the results show that GNRF has the same complexity as GCN.
>
> [1] Curvdrop: A ricci curvature based approach to prevent graph neural networks from over-smoothing and over-squashing.
>
>  [2] Curvature Graph Neural Network.
>
> [3] Hyperbolic variational graph neural network for modeling dynamic graphs
>
> [4] Large Scale Learning on Non-Homophilous Graphs: New Benchmarks and Strong Simple Methods

---

> ### Author Response · Authors · 2024-11-19
> **(2/N)**
>
> ## Question 1
> For this question, you can refer to our discussion on the differences between GNRF and GRAND in the paper. GRAND is a classic continuous-depth GNN model that directly uses attention coefficients as aggregation weights. We summarize the main difference as follows: a significant distinction lies in the sign of the aggregation weights. Attention coefficients are typically positive, whereas GNRF allows for negative weights. Specifically, when an edge has positive curvature, we use negative weights, and vice versa for negative curvature. This leads to completely different behavior between GNRF and GRAND. GRAND (attention) tends to smooth all node pairs, while GNRF only smooths node pairs with negative curvature and repels those with positive curvature. In our ablation study, we analyzed the impact when GNRF and GRAND differ only in the aggregation weights. The results showed that GRAND performed poorly on heterophilious graphs (e.g., Tolokers and Roman-Empire), which supports our view on the importance of negative weights/attention coefficients. You may further ask what would happen if we introduced the ability to use negative weights in the attention mechanism—this is precisely what another model, ACMP, does. We also provided a detailed comparison in the paper, and the experimental results show that ACMP still performs significantly worse than GNRF.
>
> ## Question 2
> There may be some ambiguity in our description of EdgeNet, which led to a misunderstanding, and we apologize for that. For datasets commonly used in graph deep learning (such as the node classification datasets we used), attributes are often only present on nodes, not edges. At each time step, we first generate an attribute for each edge (specifically, on edge i~j, we use h_i(t) || h_j(t) for concatenation). Then, in that time step, we obtain the aggregation weights through several layers of EdgeNet. The edge attributes are cleared after that time step, and the process is repeated in the next time step. Therefore, Equation 13 (which has been moved to the appendix in the updated paper) describes how to **obtain aggregation weights using a multi-layer network within a single time step**.
>
> ## Question 3
> In Appendix C.1, we added an explicit update formula for GNRF under forward differentiation method. Please note that this is only for illustration. In actual practice, the update formula for features is more complex due to our use of a more advanced ODE solver. In the field of deep learning, we focus more on describing a novel partial differential equation without discussing the internal workings of the ODE solver in detail, which is consistent with almost all related works on GNNs based on differential equations, such as [4] and [5]. Nevertheless, we highly value your feedback and will provide pseudocode in the next version of the paper.
>
> ## Question 4
> In fact, calling GNRF a single "layer" is inaccurate. In the paper, we refer to it as "continuous depth," and in the code, we call it a "block." We avoid using the term "layer." The GNRF implementation in the code has only one ODE **block**. However, it is important to note that an ODE block can simulate arbitrary depth (or, less rigorously, any number of layers) of a GNN by appropriately setting the evolution end time T. For example, if the first ODE block evolves the system from T = t_0 to T = t_1, and the second ODE block continues to evolve the system from T = t_1 to T = t_2, this is essentially equivalent to using a single ODE block to evolve the system from T = t_0 to T = t_2. Within the same ODE block, the feature update fomula is executed multiple times (the specific number and manner of execution are determined by the ODE solver and are not explicitly shown in the code). Therefore, the answer is yes—a single ODE block can effectively approximate any multi-layer discrete-depth GNN; we only need to increase the termination time T.
>
> We hope this response effectively addresses your concerns, and we are more than willing to provide further details regarding any other questions you may have and to update the paper accordingly. Thank you once again for your valuable feedback and suggestions!
>
> [5] GRAND: Graph Neural Diffusion
>
> [6] ACMP: Allen-Cahn Message Passing for Graph Neural Networks with Particle Phase Transition

---

> ### Author Response · Authors · 2024-11-22
>
> Dear Reviewer, we have completed all the planned revisions as scheduled. Specifically, we have supplemented the content with graph classification tasks (see Table 6 in Appendix C.2), and conducted classification and regression tasks on two datasets each containing over one million nodes (also documented in Table 7 of Appendix C.2). We are eagerly looking forward to your positive feedback.

---

> ### Author Response · Authors · 2024-11-23
> **(3/N)**
>
> We present experimental data here that may address your concerns for your review.
>
> ## Table 3
> We first present the experimental results for OGBN-Arxiv and OGBN-Year. We observe that, when maintaining the same depth, our method shows significant improvements over classical models, both in homophilious and heterophilious settings.
> | |OGBN-Arxiv|OGBN-Year |
> |---|---|---|
> |GCN(depth=4)|67.85|46.22|
> |GAT(head=3,depth=4)| 66.71|44.51|
> |ACMP(depth=4)|67.16|47.55|
> |GNRF(depth=4)|69.25|48.55|
>
> Next, we report the parameter count, storage, and average runtime per epoch based on OGBN-Arxiv. We extract the scenario from Table 3 where the depth is set to 64. In this depth setting, scalability becomes a significant challenge for the model.
>
> | |#Param|Mem.|Time|
> |---|---|---|---|
> |GCN|273k|12.9k|0.93s|
> |GAT|OOM|N/A|N/A|
> |ACMP|19.5k|7.15G|17.6s|
> |GNRF|35.9k|11.5G|0.79s|
>
> In the main text, we explained that GNRF has computational complexity comparable to that of GCN. However, in this experiment, we found that discrete-depth GNNs (GCN/GAT) require different parameters for each layer, causing their parameter count to increase linearly with the number of layers. In contrast, GNRF and ACMP, as continuous-depth GNNs, maintain a constant number of parameters regardless of depth. Additionally, thanks to GNRF's use of a fixed-step ODE solver, it is significantly faster than ACMP, which uses an adaptive-step solver, when facing long-duration evolution processes. As a result, GNRF achieves a favorable balance across various resource consumption metrics.
>
> ## Table 7
> We conducted experiments on two graph task datasets with over 1 million nodes. These datasets are highly challenging for general message-passing GNNs and are often used to validate the adversarial robustness of models under over-squashing. The results show that our method significantly outperforms GCN, and even competes with SAN (a Graph Transformer-based model with much higher complexity than GNRF). This also demonstrates that GNRF is well-suited for large-scale datasets.
>
> |           | GCN                | GatedGCN+RWSE       | SAN+LapPE           | SAN+RWSE            | GNRF                | GNRF+LapPE          | GNRF+RWSE           |
> |-----------|--------------------|---------------------|---------------------|---------------------|---------------------|---------------------|---------------------|
> | Peptides-func AP(↑) | 0.5930±0.0023      | 0.6069±0.0035       | 0.6384±0.0121       | 0.6439±0.0075       | 0.6233±0.0080         | 0.6455±0.0062       | 0.6480±0.0056       |
> | Peptides-struct MAE(↓) | 0.3496±0.0013      | 0.3357±0.0006       | 0.2683±0.0043       | 0.2545±0.0012       | 0.3166±0.0053       | 0.2675±0.0044       | 0.2811±0.0031       |

---

> ### Author Response · Authors · 2024-11-24
> **(4/N)**
>
> Dear reviewer, we agree with your point in Weakness 1 that "it would be valuable to discuss the potential benefits of using more complex GNN architectures". Therefore, we have added a new section, **"Future direction" in the appendix C.3** to explore the possibility of applying our proposed Attri-DRF to Graph Transformer, another commonly used GNN architecture. We excerpt the original text for you as follows:
>
> "In this paper, we focus on the application of Attribute Discrete Ricci flow in message passing-based discrete/continuous-depth GNNs. However, in view of the excellent performance of Graph Transformers (GTs), especially on graph-level tasks, we believe that it is also meaningful to consider the application of Attri-DRF in this type of method. We believe that this generalization may be feasible, based on two observations:
>
> (1) In theory, there are certain curvatures that can be defined on any node pair $(i, j)$ without requiring $i$ and $j$ to be adjacent (For example, Ollivier Ricci curvature). This is in GTs is very useful because GTs directly aggregates information from the entire graph.
>
> (2) In practice, a significant difference between our model GNRF and GARND is that the aggregation weight replaces the attention coefficient with a curvature-aware coefficient. Since attention is widely used in GTs, this replacement is likely natural.
>
> We also provide a possible promotion here. Let $\mathsf{PE}(\cdot)$ be some position encoding function and $\mathsf{sim}(\cdot,\cdot)$ be some similarity function. We can let $w_{ij}(t) \equiv \mathsf{sim}(\mathsf{PE}(i,t), \mathsf{PE}(j,t))$ to get the generalization of Attri-DRF:
>
> $$
> \frac{\partial \mathsf{sim}(\mathsf{PE}(i,t), \mathsf{PE}(j,t))}{\partial t} = -\kappa_{ij}(t)\mathsf{sim}(\mathsf{PE}(i,t), \mathsf{PE}(j,t)).
> $$
>
>
> We leave application on GTs of this definition to future work."

---

> ### Comment · Reviewer_tPnw · 2024-11-25
> **Reply to the authors**
>
> I have read the revised paper and your responses—thank you for your hard work in addressing my concerns. I find that my previous questions and the weaknesses have been well addressed. But I have two additional suggestions for the revised paper, and I will appreciate it if you can consider them.
> * **Hyperparameters for Baseline GCN and GAT Models.**
> Would you mind sharing the hyperparameters used for the baseline GCN and GAT models? The performance reported on the OGB official website for GCN on the Ogb-Arxiv dataset is ranked 64, with a validation accuracy of 0.7300 ± 0.0017 and a test accuracy of 0.7174 ± 0.0029. It is a little different compared to the reported results in the revised paper. Providing the hyperparameters and clarifying the reasons for any differences in results (e.g., whether residual connections were used) would enhance the credibility of your results and help readers better understand the differences.
>
> * **Suggestions for the Efficiency Comparison in Table 3.**
> When comparing the efficiency of the proposed method with the baseline GCN in Table 3, GNRF requires more time for continuous-depth 4 (0.79s vs. 0.17s). In my opinion, both models are relatively fast. However, for deeper networks, the performance of both GNRF and GCN tends to degrade, which highlights another perspective on the over-smoothing issue in GNNs. Therefore, I believe it may not be suitable to include comparisons with deeper cases here.
> While such efficiency comparisons do demonstrate the training time of proposed GNRF will be the same with increasing depth, GNRF does not show performance improvements with increasing depth. Therefore, focusing the efficiency study on 4-layer networks might be sufficient. Additionally, if deeper networks are to be compared, models like DeeperGCN, which can maintain or improve performance with increasing depth, might be more appropriate for this context.
>
> By the way, I delete the minor issue about GNN training cost, since the training time complexity for GNN full batch training is linearly increased with the number of layers shown in LADIES[1]. Sorry for any inconvenience.
>
> [1] Layer-Dependent Importance Sampling for Training Deep and Large Graph Convolutional Networks
> .

---

> ### Author Response · Authors · 2024-11-25
> **(5/N)**
>
> Thank you for your response. We are glad to see that our previous efforts have clarified most of your concerns. Regarding your additional suggestion, we continue to respond as follows:
>
> + Hyper-parameters for GCN and GAT: We checked the hyperparameter settings on the OGB website and found two potential hyperparameter differences that may affect performance. First is the hidden layer size. We set the hidden layer size to 64, while OGB uses 256. As noted in Table 4 of the appendix, when using the OGBN-Arxiv dataset, we fixed the hidden size of GNRF to 64 (to avoid OOM). To ensure consistent model capacity, we also fixed the hidden layer size of all comparison methods to 64, but obviously, larger hidden layers generally lead to better performance. The second important parameter is the number of layers. We used 4, while OGB uses 3. We believe these two parameters have the most significant impact. Other parameters include: lr=0.001, epoch=2000, dropout=0.5. As for the design of GCN and GNN, we reviewed the OGB source code and found that they are essentially the same, with no residual connections used.
>
> + Suggestion on efficiency comparison: Yes, we agree with your point. Here, the deeper comparisons are mainly used to highlight two unique features of GNRF. First, as a continuous depth GNN, the number of parameters and memory usage of GNRF is independent of depth. Second, because GNRF uses a fixed-step solver, it is much faster than other popular continuous deep GNNs. We believe it is beneficial to show these two points to the readers. However, we also admit that GCN and GAT are not the most suitable choices for deep GNNs. Therefore we changed the comparison method here. You can see our reply (6). And we have implemented these improvements in the paper.
>
> If possible, we would appreciate your quick feedback. Thank you.

---

> ### Author Response · Authors · 2024-11-26
> **(6/N)**
>
> Dear Reviewer,
>
> We greatly appreciate your suggestions, and **we have made corresponding revisions to the latest paper**.
>
> Firstly, we moved the original Table 3 to the appendix (it is now Table 8), and created a new Table 3. In this new table, we modified the originally uniform model depth from 4 to 3, and then presented the results for hidden layer sizes of 16, 64, and 256, respectively. The advantage of this setting is that when the hidden layer size is 256, the model capacity of GCN aligns with the official recommendations from OGB, ensuring fairness in comparison. Based on this new table, we believe that GNRF still achieves a meaningful trade-off between efficiency and performance. We have extracted the table for your reference below:
>
>
> | **Model** | **#Param** | **Time** | **Acc.** | **#Param** | **Time** | **Acc.** | **#Param** | **Time** | **Acc.** |
> |-----------|------------|----------|----------|------------|----------|----------|------------|----------|----------|
> |        |  #Hidden=16     | #Hidden=16   |  #Hidden=16   | #Hidden=64      |#Hidden=64    | #Hidden=64    | #Hidden=256      | #Hidden=256    | #Hidden=256|
> | GCN(Depth=3)      | 3.15k      | 0.12s    | 60.95    | 15.2k      | 0.14s    | 68.55    | 110k       | 0.21s    | **71.65** |
> | GAT(Depth=3)       | 15.0k      | 0.17s    | 59.52    | 86.8k      | 0.25s    | 64.39    | 788k       | OOM      | OOM      |
> | ACMP(Depth=3)      | 4.18k      | 3.29s    | 61.03    | 32.0k      | 6.35s    | 68.89    | 374k       | OOM      | OOM      |
>  | GNRF(Depth=3)      | 5.50k      | 0.31s    | **62.11** | 52.6k      | 0.78s    | **69.33** | 701k       | OOM      | OOM      |
>
>
> In addition, in Table 8 (the original Table 3), we deleted GCN and GAT and added APPNP and GCNII (both are deep GNNs) to ensure that our discussion in this scenario is meaningful. Our observation is similar to the original one. When the depth is relatively shallow (4 or 16), GNRF still has advantages over APPNP and GCNII, but as the depth deepens, the performance of GNRF declines.
>
> We hope this meets your suggestions well and look forward to further discussions.

---

> > ### Comment · Reviewer_tPnw · 2024-11-26
> > **Reply to the authors**
> >
> > Thank you for your efforts in revising and addressing my concerns in the paper. I have updated my score to 6 to support the acceptance.

---

> > > ### Author Response · Authors · 2024-11-26
> > >
> > > We are grateful that our efforts have finally been recognized by you. We will continue to work to improve the quality of our papers.

---

### Official Review · Reviewer_tn6E · 2024-11-08

**Soundness:** 3
**Presentation:** 2
**Contribution:** 3
**Rating:** 6
**Confidence:** 2

**Summary:**

The paper introduces the dynamical system Attribute Discrete Ricci Flow (Attri-DRF) and incorporates this to propose the Graph Neural Ricci Flow (GNRF), a curvature-aware continuous GNN. This ensures that the graph Dirichlet energy can be bilaterally bounded and that the curvature decay to 0 independent of data. Using an auxiliary network (EdgeNet), the model can theoretically incorporate different types of curvature definition. GNRF has excellent performance on many data sets against a variety of discrete and continuous GNNs.

**Strengths:**

Theoretically, the paper provide several interesting results.
1. Section 3 provides guarantees on the curvature decay rate and the stable curvature limit of Attri-DRF when certain conditions are met, along with providing a bound on the Dirichlet energy when the curvature stabilizes. This indicates it may be able to avoid over-smoothing/over-squashing.
2. Incorporating recent results, the paper uses an auxiliary network (EdgeNet), which is capable of approximating arbitrary edge curvature with high precision.

Experimentally, the paper performs well on a variety of popular node classification tasks against a number of old and new discrete and continuous GNN architectures. Section 5.2 and 5.1 provides good evidence that theoretical guarantees hold in reality.

**Weaknesses:**

1. It is not clear to the reviewer how the theoretical results tie together/what assumptions are made at each step of the way.
2. The design of EdgeNet is glossed over within the paper, with only a few formulas mentioned within either the main paper or the appendix to explain it. There's also no comparison between EdgeNet's curvature values compared against any other type of curvature that it supposedly can approximate.
3. The datasets used in the experiments are relatively small datasets.

**Questions:**

1) Why does having a data-independent curvature decay rate a good thing?
2) Where does equation (3) come from? I checked the Ollivier paper and I can't find this equation there.
3) Does GNRF satisfy the theoretical results in Section 3. If it does, it would be great if the authors can clarify this a bit more within the paper.
4) Over-smoothing and over-squashing are problems caused by the message-passing design in GNNs. Considering that GNRF does not strictly adhere to the message passing design, is it appropriate to mention these problems in this work?
5) Does GNRF work on larger datasets?

---

> ### Author Response · Authors · 2024-11-18
>
> Dear Reviewer,
>
> Thank you for your professional and detailed review of our paper. We are very pleased to see your recognition of our work, and we have carefully noted your valuable comments. Below, we provide a detailed response to your feedback.
>
> ## Weakness 1
>
> In the latest version of the paper, we have added a more detailed formal version for each theorem, including all necessary assumptions in the statement. These improvements can be found in the appendix section. The main text retains an informal version that is more intuitive and easy to understand, to allow readers to quickly grasp the key conclusions.
>
> ## Weakness 2
>
> We have added detailed descriptions of the models used in the experiments in Section C.1 of the appendix, including a description of EdgeNet. Additionally, we introduced two GNRF variants in Section 5.1 of the main text, which do not use EdgeNet but instead rely on explicit curvature calculations. Our comparisons reveal that specific curvature definitions may not always perform well across different datasets (as reflected in experiments from other papers such as [1] and [2]). Therefore, using EdgeNet for adaptive curvature appears to be a better choice.
>
> ## Weakness 3
>
> We added five new datasets to the main experiments, with the largest containing about 50,000 nodes. Furthermore, the resource consumption experiments include two even larger datasets (with over 100,000 nodes). The experimental results demonstrate that our model still achieves consistent improvements on these datasets.
>
> ## Question 1
>
> We have provided a more detailed explanation of this point in the main text, covering two aspects: 1. It serves as an extension of the theoretical results. Lemma 1 describes the state of Attri-DRF "when reaching equilibrium," while Theorem 3 further explains "whether equilibrium can be reached." 2. On a practical level, it ensures consistency in the evolution process, meaning that within a finite time, all edges evolve sufficiently, ensuring synchronized evolution of the overall graph structure without worrying about parts of the graph being insufficiently developed.
>
> ## Question 2
>
> Ollivier's paper does not directly present this equation, primarily because Ollivier's work is early research, and its notation differs somewhat from today's conventions. However, Ollivier indeed first explored a dynamic process very similar to curvature flow in his paper. Other related works also adopt a similar perspective to ours, recognizing that Ollivier's paper was the first to introduce Ricci flow on graphs (e.g., see page 5 of [3]).
>
> ## Question 3
>
> We highly value the consistency of GNRF with the theoretical results. In Sections 5.2 and 5.3 of the updated paper, we conducted detailed experiments to investigate this. The results show that GNRF does align well with the theory, including properties like curvature approaching zero (Lemma 1), uniform decay (Theorem 3), and bounded energy (Theorem 2). We appreciate your feedback and will clarify this point further in the paper.
>
> ## Question 4
>
> Indeed, the issues of over-smoothing and over-squashing were first raised in the context of discrete-depth GNNs. However, as we have added in Section 3 of the updated paper, classic continuous-depth GNNs (such as GRAND) fully adhere to the design principle of heat diffusion, one of whose fundamental characteristics is reaching thermal equilibrium, i.e., nodes becoming completely uniform. This is consistent with the concept of over-smoothing, and we have validated this in the experiments presented in Section 5.3 of the updated paper. Regarding the over-squashing problem, to the best of our knowledge, there has not yet been dedicated research on this challenge in the context of continuous-depth GNNs. However, as stated in the paper, we have found that many current methods aimed at solving the over-squashing problem share a striking consistency: reducing the influence of edges with extreme positive/negative curvature. We also note that these methods typically treat curvature as a static, topology-dependent attribute. While our approach is conceptually similar to these methods, the way we utilize curvature is entirely different, and we believe this offers a new perspective for formally addressing this challenge in the future.
>
> ## Question 5
>
> Based on the experiments added in Section 5.1 of the main text, GNRF has proven effective even on larger datasets. Moreover, we observed that GNNs based on differential equations often demonstrate stronger advantages when dealing with very large-scale model settings.
>
> We hope that our responses effectively address your questions, and we are very willing to provide more detailed replies to any further questions you may have and promptly update the paper. Thank you again for your valuable comments!
>
> [1] Curvature filtrations for graph generative model evaluation
> [2] Curvature constrained mpnns: Improving message passing with local structural properties
> [3] Graph Pooling via Ricci Flow

---

> > ### Comment · Reviewer_tn6E · 2024-11-21
> > **Response 1**
> >
> > Thank you for responding to my review.
> >
> > Regarding weakness 3, where's the performance comparison on the OBGN datasets?
> >
> > Regarding question 2, what is the dynamic process similar to curvature flow that you mentioned? I can't find this in Ollivier's paper either. I don't yet see how this 'curvature' equation is justified.
> >
> > Regarding question 4: Let me clarify my question. Over-smoothing and over-squashing are caused by the discrete/continuous message-passing design in GNNs. The curvature is just a proxy to measure how connected/bottlenecked the graph is, which is only relevant since message passing utilizes the graph topology to propagate information. The curvature itself doesn't have a direct relevant to the learning task. Considering that GNRF does not strictly adhere to the message passing design, is it appropriate to mention these problems in this work?

---

> ### Author Response · Authors · 2024-11-22
>
> ## Weakness 3
> The results for OGBN-Arxiv and OGBN-Year are reported in **Table 3 (resource consumption experiments)**. We focus on the scalability of RNRF on these larger datasets while also reporting performance. Below is an excerpt from the table (where $d$ denotes the depth of the model):
>
> |Dataset|GCN(d=4)|GCN(d=16)|GCN(d=64)|GAT(d=4)|GAT(d=16)|GAT(d=64)|ACMP(d=4)|ACMP(d=16)|ACMP(d=64)|GNRF(d=4)|GNRF(d=16)|GNRF(d=64)|
> |---|---|---|---|---|---|---|---|---|---|---|---|---|
> |OGBN-Arxiv|67.85|56.81|33.09|66.71|OOM|OOM|67.16|65.72|51.72|69.25|65.14|55.23|
> |OGBN-Year|46.22|42.94|38.01|44.51|OOM|OOM|47.55|43.53|42.31|48.55|44.13|40.15|
>
> ## Question 2
> We referenced Ollivier's paper [1] in our work, where Ollivier defined the Coarse Ricci curvature (now known as Ollivier-Ricci Curvature, abbreviated as ORC). Subsequently, [1] introduced a continuous-time version of ORC defined as follows:
> $$
> \kappa(x,y) = -\frac{d}{dt}\frac{W_1(m_x^t,m_y^t)}{d(x,y)}
> $$
>
> Here, $m_x^t$ epresents the probability distribution of a random walk at point $x$ at time $t$. Ollivier also discussed how this formula could be extended to graphs by treating $m_x^t$ as the probability distribution of a random walk starting at node $x$ and transitioning to its first-order neighbors (with probabilities determined by edge weights).
> Please note that this actually contains the idea of ​​Ricci flow: edge weights change over time, making $x$ time-dependent and, consequently, curvature time-dependent.
>
> A few months later, Ollivier significantly expanded upon this in his paper (2010, [2]). In Section 2.3.5, “Problem N,” Ollivier formally proposed Discrete Ricci flow, using the following equation:
> $$
> \frac{d}{dt}d(x,y)=-\kappa(x,y)d(x,y)
> $$
> This equation is nearly identical to the one we used. In [2], Ollivier explained that this equation was inspired by results in continuous Riemannian geometry. The first application of this formula in graph learning was in [3]. Our innovation lies in applying Discrete Ricci flow to other curvature definitions.
>
> It is worth noting that Discrete Ricci flow had already been widely used in the field of computer graphics before [3]. For example, in [4], the following equation was used:
> $$
> \frac{dg_{ij}(t)}{dt}=-2K(t)g_{ij}(t)
> $$
> Here, $g_{ij}(t)$ is a distance metric on the manifold, and $K(t)$ is the corresponding Gaussian curvature. Although [4] did not mention Ollivier’s work, the formulas are formally identical. We will add citations to the above works, especially [2], in our paper to avoid confusion for readers.
>
> ## Question 4
> We respectfully offer a different perspective on this matter. We believe that GNRF is, in fact, a fully message-passing framework. Specifically, the equation we used in the paper is as follows:
>
> $$
> \frac{\partial\boldsymbol{h}_i(t)}{\partial t} = \sum -{\rm EdgeNet}(t) [\boldsymbol{h}_j(t) -  {\cos\big(\boldsymbol{h}_j(t), \boldsymbol{h}_i(t)\big)}\boldsymbol{h}_i(t)]
> $$
>
> Using the simplest ODE solver (i.e., the forward Euler method), we derive the following explicit update process:
>
> $$
> {\boldsymbol{h}_i(t+1)} = \boldsymbol{h}_i(t) - \eta\sum -{\rm EdgeNet}(t) [\boldsymbol{h}_j(t) -  {\cos\big(\boldsymbol{h}_j(t), \boldsymbol{h}_i(t)\big)}\boldsymbol{h}_i(t)]
> $$
>
> This formula fully aligns with the three-stage message-passing paradigm—Message, Aggregation, and Update:
>
> Message function:
>
> $$
> M_{ij}(t) = \boldsymbol{h}_j(t) -  {\cos\big(\boldsymbol{h}_j(t), \boldsymbol{h}_i(t)\big)}\boldsymbol{h}_i(t)
> $$
>
> Aggregate function:
>
> $$
> h^\prime_i(t) = \sum -{\rm EdgeNet}(t)M_{ij}(t)
> $$
>
> Update function:
> $$
> h_i(t+1) = h_i(t) - \eta h^\prime_i(t)
> $$
>
> More advanced ODE solvers only modify the Update function. Therefore, GNRF still entirely fits within the message-passing framework.
>
> We sincerely hope this addresses your concerns.
>
> [1] Ricci curvature of markov chains on metric spaces.
>
> [2] A survey of Ricci curvature for metric spaces and Markov chains
>
> [3] Network Alignment by Discrete Ollivier-Ricci Flow
>
> [4] Discrete Surface Ricci Flow

---

> ### Author Response · Authors · 2024-11-22
>
> Dear Reviewer, We have now completed all the planned revisions. Specifically, we have added more references related to discrete Ricci flow in the main text; and introduced more diverse and larger datasets, which are detailed in Appendix C.2, Tables 6 and 7. We are eagerly awaiting your positive response.

---

> > ### Comment · Reviewer_tn6E · 2024-11-24
> > **Response 2**
> >
> > I thank the authors for their responses. I think this paper presents an interesting idea, and would like to keep my score as is.

---

> > > ### Author Response · Authors · 2024-11-24
> > >
> > > We are very grateful to the reviewers for their seriousness and responsibility, and we are happy to see that you recognized our work. We will continue to work hard to continuously improve the quality of this paper.

---

### Author Response · Authors · 2024-11-18
**A Better version now available!**

Sorry to keep the reviewers waiting so long! We highly valued your professional comments and revised our paper comprehensively, which took a couple of days because the workload was a bit much. Specifically, our revisions are as follows:

# More rigorous statement of the theorem
1. We add a new formal version of all theorems in the appendix that is more detailed than before, and retain the more intuitive and accessible informal version in the main text (tn6E, QrP8)

# More in-depth discussion
1. We add a detailed design description of EdgeNet in the appendix section (tn6E, tPnw) .
2. We add a description of the computational complexity of the algorithm in the main text (tPnw)
3. We further explain the advantages of data-independent decay rates in the main text (tn6E)
4. We make the differences with GRAND more explicit in Section 4.1 (QrP8)
5. We analyze more work related to graph curvature, Riemannian graph learning (tPnw)
6. We further discuss the advantages of using EdgeNet to approximate curvature in Section 4.2 (tn6E, QrP8)
7. We re-organize the formulation of the ablation study to illustrate the differences with existing work such as GRAND (QrP8 )

# Richer experiments
1. We report results on larger datasets (OGBN-Arxiv and OGBN-Year) (tn6E, tPnw, 1v62), along with the number of trainable parameters, the peak memory footprint, and the average single-round training time (tPnw)
2. The main experiment includes a larger dataset and a more strong baseline, while using more rational evaluation metrics (e.g., ROC-AUC in Tolokers' evaluation) (tn6E, 1v62)
3. We perform ablation experiments for the case where real curvature is used instead of EdgeNet (tn6E, QrP8)

We are eager to discuss the current updated version with reviewers as soon as possible, and we are willing to continue to make rapid adjustments to the paper in response to further feedback.

---

### Author Response · Authors · 2024-11-22
**The second modified version is now available!**

We have conducted a second comprehensive revision of the paper, incorporating all planned changes. Specifically, these include:

1. **Supplementing Missing References**: Reviewer tn6E raised concerns about the unclear origin of discrete Ricci flow. We have now provided more relevant references.

2. **Adding Pseudocode**: Reviewer tPnw suggested including pseudocode. We have addressed this by adding pseudocode in Appendix C.1.

3. **Providing Additional Experiments**: Reviewer 1v62 believed that our experimental results were limited. In response, we have included additional experiments in Section C.2. Table 6 presents the performance of GNRF on three commonly used graph classification datasets, while Table 7 evaluates GNRF on long-range graph benchmarks. These benchmarks involve datasets with over one million nodes, and the results demonstrate the strong performance of our method.

4. **Future Direction**: Reviewer tPnw thought it would be beneficial to discuss the application of our framework to a wider range of GNNs. We now show in Appendix C.3 an intuition of how to apply Attri-DRF on Graph Transformer. We also show why this intuition is reasonable.

**Additional Changes**:
5. We have further developed Theorem 2 by providing additional proof that the Dirichlet energy lower bound obtained in Theorem 2 is strictly greater than zero. This ensures that our conclusions are non-trivial.

6. We highlighted all theorems to make the paper look better.

---

### Meta-Review · Area_Chair_bxbX · 2024-12-22

**Metareview:**

In the paper, the authors introduce the dynamical system Attribute Discrete Ricci Flow (Attri-DRF) and incorporate it into a novel framework called Graph Neural Ricci Flow (GNRF), a continuous graph neural network that is curvature-aware.

After the rebuttal, most of the concerns were addressed. There are several strengths of the current paper: (1) The proposed framework is novel and interesting. Theoretically, the results are sound and solid (e.g., guarantees on the curvature decay rate and the stable curvature limit of Attri-DRF in Section 3).

While there are still some concerns about limited experiments and evaluations, in my opinion the strengths outweigh the weaknesses. As a consequence, I recommend accepting the paper. The authors are encouraged to incorporate the suggestions and feedback of the reviewers into the revision of their manuscript.

**Additional Comments On Reviewer Discussion:**

Please refer to the metareview.

---

### Decision · Program_Chairs · 2025-01-22

Accept (Poster)